



# The impact of RCM formulation and resolution on simulated precipitation in Africa

Minchao Wu[1], Grigory Nikulin[1], Erik Kjellström[1], Danijel Belušić[1], Colin Jones[2] and David Lindstedt[1]

Correspondence to: Minchao Wu (minchaowu.acd@gmail.com)

[1]Swedish Meteorological and Hydrological Institute, Folkborgsvägen 17, 60176 Norrköping, Sweden

[2]National Centre for Atmospheric Science (NCAS), University of Leeds, Leeds, UK





# Abstract

We investigate the impact of model formulation and horizontal resolution on the ability of

Regional Climate Models (RCMs) to simulate precipitation in Africa. Two RCMs - SMHI-RCA4

and HCLIM38-ALADIN are utilized for downscaling the ERA-Interim reanalysis over Africa at

four different resolutions: 25, 50, 100 and 200 km. Additionally to the two RCMs, two different

configurations of the same RCA4 are used. Contrasting different RCMs, configurations and

resolutions it is found that model formulation has the primary control over many aspects of the

precipitation climatology in Africa. Patterns of spatial biases in seasonal mean precipitation are

mostly defined by model formulation while the magnitude of the biases is controlled by

resolution. In a similar way, the phase of the diurnal cycle is completely controlled by model

formulation (convection scheme) while its amplitude is a function of resolution. Although higher

resolution in many cases leads to smaller biases in the time mean climate, the impact of higher

resolution is mixed. An improvement in one region/season (e.g. reduction of dry biases) often

corresponds to a deterioration in another region/season (e.g. amplification of wet biases). The

experiments confirm a pronounced and well known impact of higher resolution - a more realistic

distribution of daily precipitation. Even if the time-mean climate is not always greatly sensitive

to resolution, what the time-mean climate is made up of, higher order statistics, is sensitive.

Therefore,  the realism of the simulated precipitation increases as resolution increases.

Our results show that improvements in the ability of RCMs to simulate precipitation in Africa

compared to their driving reanalysis in many cases are simply related to model formulation and





not necessarily to higher resolution. Such model formulation related improvements are strongly
model dependent and in general cannot be considered as an added value of downscaling.

Keywords: RCA4, HCLIM, Resolution dependency, Added value, CORDEX-Africa


















# 1 Introduction

Regional climate modeling is a dynamical downscaling method widely used for downscaling

coarse-scale global climate models (GCMs) to provide richer regional spatial information for

climate assessments and for impact and adaptation studies (Giorgi and Gao, 2018; Giorgi and

Mearns, 1991; Laprise, 2008; Rummukainen, 2010). It is well-established that regional climate

models (RCMs) are able to provide added value (understood as improved results) compared to

their driving GCMs. This includes better representation of regional and local weather and climate

features as a result of better capturing small-scale processes, including those influenced by

topography, coast lines and meso-scale atmospheric phenomena (Flato et al., 2013; Prein et al.,

2016). However, added value from RCMs may have different causes and it may not always be

for the right reason where "right reason" would result from an improved representation of

regional process at smaller scales. Such improvement leads to more accurate results on local

scales, and can, to some extent, also reduce large-scale GCM biases (Caron et al., 2011;

Diaconescu and Laprise, 2013; Sørland et al., 2018). Contrastingly, added value may be

attributed to the "wrong reason", not directly related to higher resolution in RCMs but to

different model formulation in the RCMs and their driving GCMs. It is possible that the physics

of a RCM has been targeted for processes specific to the region it is being run for, giving it a

local advantage over GCMs that may have had their physics developed for global application.

However, it is questionable if improvements of such "downscaling" via physics can be

considered as an added value.  In general, RCMs can either reduce or amplify GCM biases

sometimes even changing their signs (Chan et al., 2013).



Issues as those mentioned above, have raised substantial concerns among regional climate
modelers (e.g., Castro, 2005; Xue et al., 2014). It has been pointed out that understanding of the
added value remains challenging. It would become even more complicated taking into account
the effects of different realizations, such as the size of domain, lateral boundary conditions,
geographical location, model resolution and  its internal variability (Di Luca et al., 2015; Hong
and Kanamitsu, 2014; Rummukainen, 2016). All the above factors potentially influence
downscaled results leading to different interpretation of the downscaling effects, thus the
robustness of added value. For example, it was shown that over the Alps, downscaling with
multiple RCMs at increasing resolutions in general is able to provide a more realistic
precipitation pattern than the forcing GCMs, and it is regarded as added values from RCMs
(Torma et al., 2015b). Similarly, Lucas-Picher et al (2017) found added value over the Rocky
Mountains, another region with strong topographic influence on hydrological processes.
However, the results are not unambiguous and sometimes limited added value is found when
comparing to the forcing data, (e.g. Wang and Kotamarthi, 2014) over North America. This
implies that the understanding of downscaling effects is context-dependent and one should
carefully interpret the downscaled results in order to detect robust added value.
Africa is foreseen to be vulnerable to future climate change, which early on inspired efforts to
employ RCMs for impact and adaptation studies (e.g. Challinor et al., 2007). Further to previous
coordinated downscaling activities over Africa as for example the African Monsoon
Multidisciplinary Analyses (AMMA) (Van der Linden and Mitchell, 2009), the Coordinated
Regional climate Downscaling Experiment (CORDEX)  provides a large ensemble of RCM
projections for Africa (Giorgi et al., 2009; Jones et al., 2011).  All CORDEX RCMs follow a





common experiment protocol including a predefined domain at 50km resolution and common
output variables and format that facilitates assessment of projected climate changes in Africa.
Under this framework, RCMs at 50-km horizontal resolution are found to have the capability of
providing added value in representing African climatological features compared to their forcing
GCMs, which generally have the resolution coarser than 100 km (Dosio et al., 2015;
Moufouma-Okia and Jones, 2015; Nikulin et al., 2012). However, a number of common
problems with the RCMs are identified, which include, for example, dry biases over
convection-dominated regions like the Congo basin, too early onset of the rainy season for the
West African Monsoon region and biases in representing the diurnal cycle of precipitation (Kim
et al., 2014; Laprise et al., 2013; e.g. Nikulin et al., 2012). So far, it is still not clear if differences
between the CORDEX Africa RCMs and their driving GCMs  are related  to higher RCM
resolution or to RCM internal formulation, or to the combination of both. A thorough
understanding of such differences and of added value of the CORDEX-Africa RCMs is
necessary for robust regional assessments of future climate change and its impacts in Africa.
In this study, we aim to separate the impact of model formulation and resolution on the ability of
RCMs to simulate precipitation in Africa. We conduct a series of sensitivity, reanalysis-driven
experiments by applying two different RCMs, one of them in two different configurations, at
four horizontal resolutions. Contrasting the different experiments allow us to separate the impact
of model formulation and resolution. We present an overview and the first results of the
experiments conducted and leave in-depth detailed process studies for different regions to
forthcoming papers.



# 2 Methods and Data

## 2.1 The Regional Climate Models

### 2.1.1 RCA4

The Rossby Centre Atmosphere regional climate model - RCA (Jones et al., 2004; Kjellström et al., 2005; Räisänen et al., 2004; Rummukainen et al., 2001; Samuelsson et al., 2011) is based on the numerical weather prediction model HIRLAM (Undén et al. 2002). To improve model transferability, the latest fourth generation of RCA, RCA4, has a number of modifications for specific physical parameterizations. This includes the modification of convective scheme based on Bechtold-Kain-Fritsch scheme (Bechtold et al., 2001) with revised calculation of convective available potential energy (CAPE) profile according to Jiao and Jones (2008), and the introduction of turbulent kinetic energy (TKE) scheme (Lenderink and Holtslag, 2004). The RCA4 model has been  applied in many regions worldwide, among them Europe (Kjellström et al., 2016, 2018; Kotlarski et al., 2015), the Arctic (Berg et al., 2013; Koenigk et al., 2015; Zhang et al., 2014), Africa (Nikulin et al., 2018; Wu et al., 2016), South America (Collazo et al., 2018; Wu et al., 2017), South-East (Tangang et al., 2018) and South Asia (Iqbal et al., 2017).

In addition to the standard RCA4 configuration, used in CORDEX, in this study we also include a RCA configuration with reduced turbulent mixing in stable situations (especially momentum mixing). Such change in model formulation was applied to reduce a prominent dry bias found in RCA4 CORDEX Africa simulations over Central Africa (Tamoffo et al., 2019; e.g. Wu et al., 2016). Using two configurations of RCA4 allows us to examine how sensitive our results are to





different formulations of the same model. We hereafter denote the original RCA4 configuration
as RCA4-v1 and the new one as RCA4-v4.
### 2.1.2 HCLIM
HARMONIE-Climate (HCLIM) is a regional climate modelling system designed for a range of
horizontal resolutions from tens of kilometers to convection permitting scales of 1-3km
(Belušić et al., 2019; Lindstedt et al., 2015). It is based on the ALADIN-HIRLAM numerical
weather prediction system (Belušić et al., 2019; Bengtsson et al., 2017; Termonia et al., 2018).
The HCLIM system includes three atmospheric physics packages AROME, ALARO and
ALADIN, which are designed for different horizontal resolutions. The ALADIN model
configuration used in this study employs the hydrostatic ARPEGE-ALADIN dynamical core
(Temperton et al., 2001), a mass-flux scheme based on moisture convergence closure for
parameterizing deep convection (Bougeault, 1985) and SURFEX as the surface scheme (Masson
et al., 2013). All details about the version of HCLIM used in this study (HCLIM38), and its
applications over different regions can be found in (Belušić et al., 2019). We need to note that
HCLIM38-ALADIN used in the study is not the same model as ALADIN-Climate used in
CORDEX (Daniel et al., 2019). We refer to HCLIM38-ALADIN as HCLIM-ALADIN hereafter.

## 2.2 Experimental design

To investigate the response of both RCA4 and HCLIM-ALADIN to horizontal resolution, we
conduct a set of sensitivity experiments driven by the ERA-Interim reanalysis (denoted as
ERAINT hereafter; Dee et al., 2011) at four different resolutions. These resolutions are 1.76,
0.88, 0.44 and 0.22° for RCA4 with the rotated coordinate system and 200, 100, 50 and 25km for

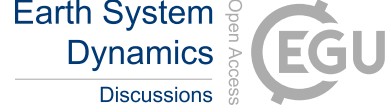

HCLIM-ALADIN with the Lambert Conformal projection. The 0.44° or 50km resolution is
recommended by the CORDEX experiment design and used in the CORDEX-Africa ensemble.
Hereafter, the resolution in kilometers is used unless otherwise specified. The setup of the
simulations at the four resolutions is identical apart from the timestep that is adjusted to ensure
numerical simulation stability and the size of the full computational domain with the relaxation
zone (see Table 1). The relaxation zone has 8 grid-points in all directions and increases (in km) at
coarser resolution while the interior CORDEX-Africa domain is the same. Larger size of the
computational domain at coarser resolution may have a potential impact on the results leading to
larger internal variability developed by the RCMs and weaker constraints on the ERAINT
forcing. We perform an additional experiment with RCA4 at 0.88° resolution taking the full
computational domain from the 1.76° RCA simulation. For precipitation differences between the
two experiments are at the noise level while for seasonal mean temperature it can be up to 1°C.
The RCA4 0.88° simulations and the HCLIM-ALADIN 100km one represent a slight upscaling
of ERAINT (about 0.7° or about 77km at the Equator) and we refer to them as "no added value
experiment". No resolution-dependent added value of the RCMs is expected for these
simulations and all differences between the RCMs and their driving ERAINT are attributed to
different model formulations. We note that in general, both regional models - RCA and
HCLIM-ALADIN were developed to operate at a range of 10-50km resolution and their
performance at 100 and 200km may not be optimal. All simulations are conducted without
spectral nudging and analysis is done for the CORDEX-Africa domain shown in Fig. 1.

Table 1. Details of the RCA4 and HCLIM ALADIN experiments





| Experiment name | Horizontal resolution (deg. / km) | Domain size (lon × lat) | Geographical area (deg.) | | Time step (sec) |
|---|---|---|---|---|---|
| | | | South, North | West, East | |
| RCA4-v* 1.76° | 1.76° | 66 × 67 | -60.5, 55.66 | -38.06, 76.34 | 1200 |
| RCA4-v* 0.88° | 0.88° | 126 × 121 | -54.78, 50.82 | -33.22, 76.78 | 1200 |
| RCA4-v* 0.44° | 0.44° | 222 × 222ᵃ | -50.16, 47.08 | -29.04, 68.20 | 1200 |
| RCA4-v* 0.22° | 0.22° | 406 × 422 | -48.07, 44.55 | -26.95, 62.15 | 600 |
| HCLIM-ALADIN 200km | 200 km | 80 × 90 | -58.34, 56.71 | -46.98, 82.98 | 1800 |
| HCLIM-ALADIN 100km | 100 km | 128 × 150 | -53.89, 51.70 | -37.01, 73.01 | 1800 |
| HCLIM-ALADIN 50km | 50 km | 240 × 270 | -51.56, 48.98 | -35.85, 71.85 | 1200 |
| HCLIM-ALADIN 25km | 25 km | 450 × 512 | -50.43, 47.73 | -33.64, 69.64 | 600 |


**CORDEX Africa | 0.44° (50 km)**

**Figure 1** Topography (m) for the the CORDEX-Africa domain in RCA4 at 50km resolution. Boxes
indicate the four subregions used for spatially averaged analysis: West Africa (WA), East Africa (EA), the
southern Central Africa (CA-S), and eastern southern Africa (SA-E).



## 2.3 Observations and reanalysis

Observational datasets in Africa, in general, agree well for large-scale climate features but can

deviate substantially at regional and local scales (Fekete et al., 2004; Gruber et al., 2000; Nikulin

et al., 2012). To take into account the observational uncertainties, we utilize a number of gridded

precipitation datasets. They include three gauged-based datasets: the Global Precipitation

Climatology Centre, GPCC, version 7 (Schneider et al., 2014), the Climate Research Unit

Time-Series, CRU TS, version 3.23 (Harris et al., 2014), and University of Delaware, UDEL,

version 4.01 (Legates and Willmott, 1990). All these three datasets are at 0.5° horizontal

resolution. For the evaluation of precipitation extremes and diurnal cycle simulated by RCMs,

we utilize a satellite-based precipitation dataset from the Tropical Rainfall Measuring Mission,

TRMM 3B42 version 7 (Huffman et al., 2007), which is at 0.25° horizontal resolution and

3-hourly temporal resolution. ERAINT as the driving reanalysis is also used for analysis. In

contrast to climate models, ERAINT precipitation is a short term forecast product and there are

several ways to derive ERAINT precipitation (e.g. different spin-up, base time and forecast

steps) that can lead to different precipitation estimates (Dee et al. 2011). ERAINT precipitation is

derived by the simplest method, without spinup as in some of the previous studies (Dosio et al.,

2015; Moufouma-Okia and Jones, 2015; Nikulin et al., 2012): 3-hourly precipitation uses the

base times 00/12 and forecast steps 3/6/9/12 hours, while daily precipitation uses base times

00/12 and forecast steps of 12 hours. The RCMs and ERAINT represent 3-hourly mean

precipitation for the 00:00-03:00, 03:00-06:00, … 21:00-00:00 intervals while TRMM

precipitation averages represent approximately the 22:30–01:30, 01:30–04:30, . . . 19:30–22:30

UTC intervals.






## 2.4 Methods

The coarsest resolution 200 km is used as a reference resolution for spatial maps. The
higher-resolution simulations are aggregated to the 200 km grid by the first-order conservative
remapping method (Jones, 1999). In this way we expect that the difference among the aggregated
results at common resolution should mainly be caused by the different treatment for fine-scale
processes (Di Luca et al., 2012). For the regional analysis, such as the analysis of annual cycle,
diurnal cycle and daily precipitation intensity, we focus on four subregions, presenting different
climate zones in Africa: West Africa (10°W~10°E, 7.5°N~15°N), East Africa (30°E~40°E,
15°S~0°S), the southern Central Africa (10°E~25°E, 10°S~0°S), and the eastern South Africa
(20°E~36°E, 35°S~22°S) as defined in Fig. 1. The period 1981-2010 is used for the analysis in
this study, unless otherwise specified.

# 3 Results and Discussion


## 3.1 Seasonal mean

In the boreal summer defined here as July-September (JAS), the tropical rain belt (TRB)
associated with the intertropical convergence zone (ITCZ) is positioned to its most northern
location with the maximum precipitation north of the Equator (Fig. 2a). CRU, UDEL and GPCC
aggregated to the 200km resolution, generally agree well with each other, with only slight local
differences (Fig. 2a-c). ERAINT overestimates precipitation over Central Africa and along the
Guinea Coast while underestimates it over West Africa, north of the Guinea Coast (Fig. 2d). All
RCA4-v1 simulations have a pronounced dry bias (Fig. 2e-h) that spatially almost coincides with





the wet bias in ERAINT and increases at coarser resolution (Fig1e-f). RCA4-v4 shows a similar
bias pattern compared to RCA4-v1 but  substantially reduces the dry bias over Central Africa at
all four resolutions (Fig. 2i-l). For both configurations of RCA4, the smallest dry bias is found at
the highest 25km resolution, although an overestimation of precipitation north of the dry bias
becomes more pronounced, especially for RCA4-v4. HCLIM-ALADIN, in general, shows some
similarities to RCA4 with a pronounced dry bias in West and Central Africa at 200km that is
strongly reduced with increasing resolution. However, a wet bias emerges on the northern flank
of the rain belt at 50 and 25km. For JAS there is a common tendency for both RCMs to generate
more precipitation at higher resolution leading to a reduction of the dry biases over Central
Africa. Such bias reduction may be considered as an resolution-related improvement. However,
the RCM simulations clearly show that the added value of higher resolution can be region
dependent. An improvement of the simulated precipitation climatology over one region
corresponds to deterioration of the climatology over another region. Moufouma-Okia and Jones
(2015) found a mixed response to resolution in simulated seasonal mean precipitation over West
Africa. Their RCM simulations at 50 and 12km bear a great deal of similarity with each other
while a simulation at 25km shows wetter conditions in the Sahel and drier ones near the coastal
area in the south (see their Fig. 8). In contrast, Panitz et al. (2014) found almost no difference in
seasonal rainfall over West Africa between two RCM simulations at 50 and 25km. We conclude
that for both RCA4 and HCLIM-ALADIN, spatial bias patterns are similar and more related to
model formulation while magnitude of biases are more sensitive to resolution. For example, the
sign of the bias pattern in our no added value RCM simulations at 100km in JAS (Fig. 2f, j, n) is
almost opposite to the sign of the bias pattern in the driving ERAINT (Fig. 2d).







**Figure 2.** GPCC7 mean JAS precipitation for 1981–2010 and differences compared to GPCC7 in (b-d) the other gridded observations, (e-h) the RCA4-v1, (i-l) RCA4-v4 and (m-p) HCLIM-ALADIN simulations.




In boreal winter (December-February, DJF), the TRB migrates to its most southerly position
covering the latitudes from southern to Central Africa, with the maximum over southern tropical
Africa and Madagascar (Fig. 3a). Similar to JAS, observational uncertainties are generally small
in DJF and there is a pronounced wet bias in ERAINT over Central Africa (Fig. 3d). At 25 and
50km RCA4-v1 has a dipole bias pattern with an underestimation of rainfall over Central Africa
and an overestimation over southern Africa. At 200km there is a pronounced deterioration in the
simulated rainfall:  a strong dry bias appears along the eastern coast and Madagascar while the
wet bias is amplified over large parts of southwestern Africa. At 25 and 50km RCA4-v4 shows a
large-scale dipole bias pattern similar in some degree to RCA4-v1. The RCA4-v4 biases are
smaller than the RCA4-v1 ones showing an impact of the re-tuning (reducing mixing in the
boundary layer). The behaviour of RCA4-v4 at coarser resolution is also similar to RCA4-v1. A
similar strong dry bias is emerging along the eastern coast at 200km. However, in contrast to
RCA4-v1, the dry bias over the Democratic Republic of Congo almost completely disappears at
both 100 and 200km. HCLIM-ALADIN simulates almost the same bias pattern at all resolutions,
strongly underestimating rainfall over southeastern Africa and overestimating it over the Guinea
Coast, parts of central Africa and southern Africa. There is a tendency to an increase in
precipitation with higher resolution in HCLIM-ALADIN: the wet biases are amplified and the
dry biases are reduced. Both RCA4 and HCLIM-ALADIN show a common feature -
intensification of the dry bias along the eastern coast of Africa at 200km. Even, if both RCMs
have this dry bias in common, there are also differences showing the importance of model
formulation. HCLIM-ALADIN has about the same bias pattern at all four resolutions while the
RCA4 bias pattern substantially changes across the resolutions. Such resolution dependency in





RCA4 may be related to the fact that RCA4 is based on a limited area model and not developed
to operate at 100-200km resolution. Contrastingly, HCLIM-ALADIN that is based on a global
model shows more consistent results even at 100-200km resolution. Although, we also note that
the resolution dependency of the RCA4 bias pattern over southern Africa is similar to that found
for the CMIP5 GCMs (Munday and Washington, 2018). They show that the GCMs with the
coarsest resolution and respectively the lowest topography have the wettest bias over the
Kalahari basin and the driest bias over the southeast Africa coast, the Mozambique Channel and
Madagascar. Such a bias pattern is related to a smoother barrier to northeasterly moisture
transport from the Indian Ocean that penetrates across the high topography of Tanzania and
Malawi into subtropical southern Africa. However, in our analysis, HCLIM-ALADIN does not
show such resolution-related dependency. In general, similar to JAS, the added value of higher
resolution in DJF is region dependent: with higher resolution biases are reduced over one region
but amplified over another.






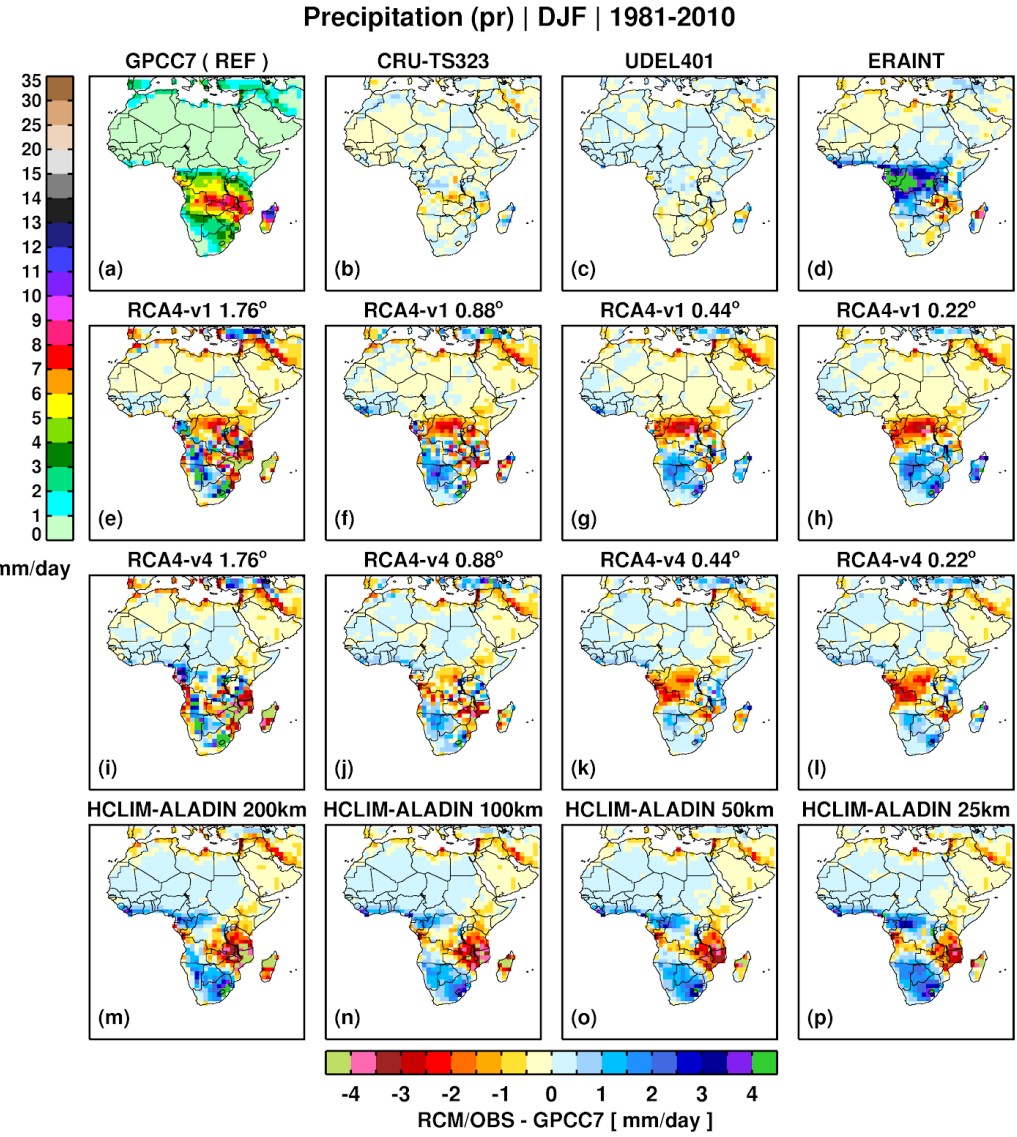

**Figure 3.** As Fig. 2, but for DJF.


## 3.2 Annual cycle

The annual cycle of precipitation over the four subregions is shown in Fig. 4. The observed
annual cycle of precipitation over West Africa depicts the West African Monsoon (WAM)



rainfall, with maximum precipitation in August (Fig. 4a). All observational datasets and
ERAINT agree well with each other with only a small underestimation of rainfall by ERAINT in
June-August. In contrast to the observations, RCA4-v1 has a bimodal annual cycle with a too
early onset of the rainy season (Fig. 4b). The simulated rainfall is overestimated in March-May,
underestimated in July-August during the active WAM period and is well in line with the
observations during the cessation of the WAM rainfall in September-November. RCA4-v4 shows
a similar behaviour but the first rainfall peak in May is reduced and the annual cycle has a more
unimodal shape (Fig. 4c). HCLIM-ALADIN, in general, shows similar features as both
configurations of RCA4, although has more similarities with RCA4-v4 (Fig. 4d). The too early
onset of the rainy season is a common problem for many RCMs reported by Nikulin et al.,
(2012). Our results show that this is not dependent on resolution but instead related to model
formulation. Higher resolution reduces the wet bias during the onset of the rainy season for
RCA-v1, has no impact for RCA-v4 and amplifies the wet bias in HCLIM-ALADIN.
Nevertheless, the impact of higher resolution is more consistent during the rainy season.
Increasing resolution tends to increase monsoon rainfall for both RCMs, resulting in smaller dry
biases and a pattern closer to the unimodal one in the observations. Eastern and Central Africa
have a bimodal annual cycle of rainfall with two peaks around November and May (Fig. 4e,i).
GPCC, CRU and UDEL agree well on the phase and magnitude of the annual cycle for both
subregions. ERAINT has a weaker bimodality overestimating precipitation in
December-February over Eastern Africa and all year round over Central Africa with the largest
wet bias during October-April. Both configurations of RCA4 fail to reproduce the bimodal
annual cycle in Eastern Africa at 200km underestimating precipitation all year round and



showing a single rainfall peak in December (Fig. 4j,k). Increasing resolution reduces the dry bias
and leads to an improvement in the shape of the annual cycle. The bimodal shape begins to
appear at 100km and becomes much closer to the observation at 50 and 25km. Despite some
mixed dry and wet biases in different seasons, the 25 and 50km RCA4 simulations show the best
agreement with the observations. In contrast to RCA4, HCLIM-ALADIN simulates the unimodal
annual cycle at all four resolutions and some sign of bimodality only appears at 25km (Fig. 4h).
Similar to RCA4, increasing resolution leads to an increase in precipitation in HCLIM-ALADIN,
although a dry bias is a prominent feature from November to May in all HCLIM-ALADIN
simulations. For Central Africa, the bimodality of the annual cycle is well reproduced by both
RCMs at all resolutions (Fig. 4j-l). An interesting feature is that RCA4 shows completely
opposite behavior in Central Africa compared to Eastern Africa. Increasing resolution leads to
decreasing precipitation for both configurations of RCA4 during the rainy seasons and especially
in January. HCLIM-ALADIN maintains similar behavior to that in Eastern Africa, although
difference in precipitation across the resolutions is small (Fig. 4l). Both RCMs strongly reduce
the ERAINT wet bias even in the no-added value experiment at 100km. Such improvement
indicates that model formulation plays a more important role than resolution over Central Africa.
For the eastern Southern Africa, the annual cycle of precipitation is unimodal with its maximum
in austral summer (Fig. 4m). Similar to West Africa, uncertainties between observational datasets
and reanalysis are small. RCA4 in general overestimates rainfall during the rainy season with the
largest wet bias at 200km. Surprisingly, the simulated rainfall is almost the same at 25 and
100km  while the smallest bias is found at 50km for both RCA4 configurations.
HCLIM-ALADIN also overestimates precipitation during the rainy season at all four resolution

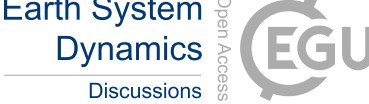



(Fig. 4p). However, the smallest wet bias in the HCLIM-ALADIN simulations is found at 50 and
100km.

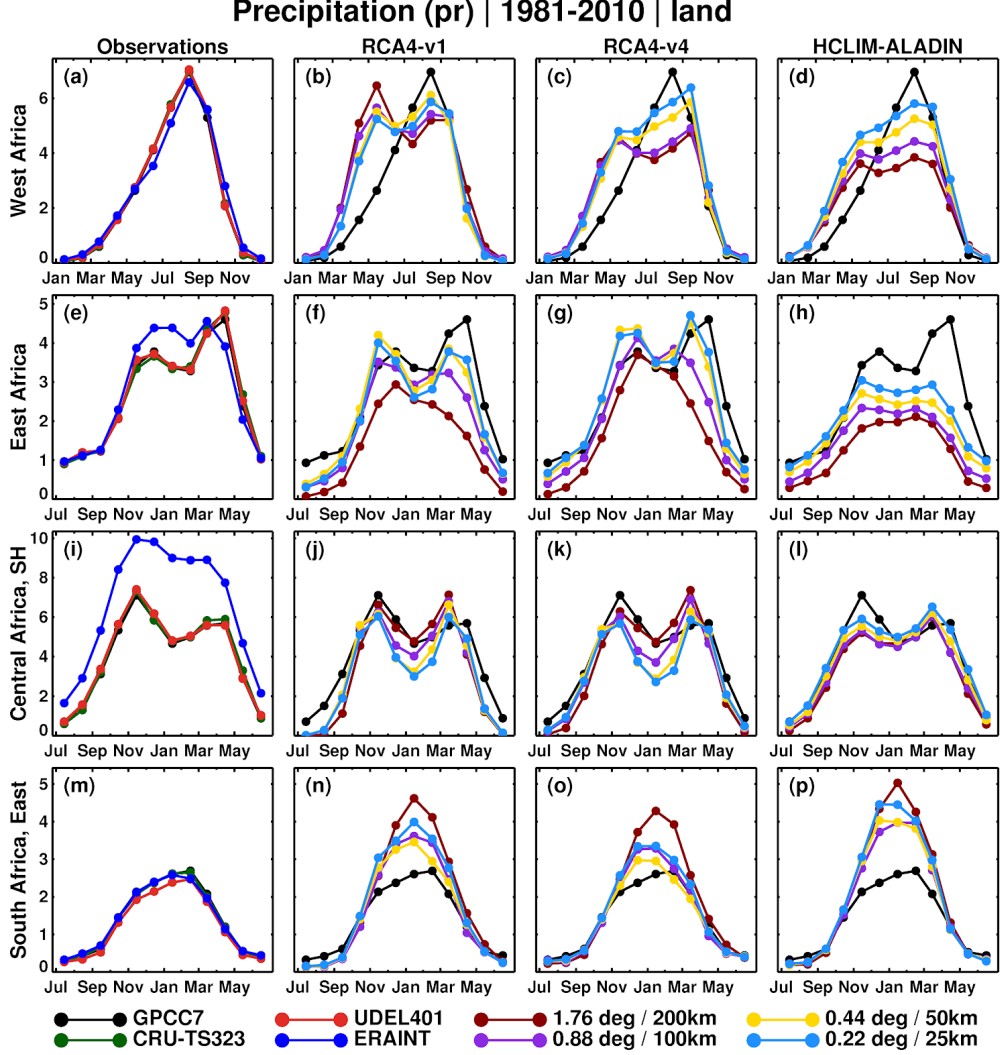

**Figure 4.** Annual cycle of precipitation over the four subregions in observations/ERAINT and as
simulated by RCA4 and HCLIM-ALADIN at the four different resolutions. Only land grid boxes are used
for averaging over the subregions. Units are mm/day.






## 3.3 Diurnal cycle

The diurnal cycle is a prominent feature of forced atmospheric variability with a strong impact
on regional- and local-scale thermal and hydrological regimes. The diurnal cycle of precipitation
in the tropics is well documented and includes a late afternoon/evening maximum over land (Dai
et al., 2007). However, it is still a common challenge for GCMs (Dai, 2006; e.g. Dai and
Trenberth, 2004; Dirmeyer et al., 2012), RCMs (e.g. Da Rocha et al., 2009; Jeong et al., 2011;
Nikulin et al., 2012) and reanalyses (Nikulin et al., 2012) to accurately  represent the diurnal
cycle of precipitation.
The TRMM diurnal cycle of precipitation generally shows an increase of rainfall starting around
the noon with maximum reached at around 18:00 local solar time (LST) (Fig. 5). The ERAINT
diurnal cycle is completely out of the phase over all subregions with the occurrence of maximum
precipitation intensity around local noon. A common feature of ERAINT is an overestimation of
precipitation around local noon and an underestimation during the rest of the day.
HCLIM-ALADIN shows exactly the same behaviour as ERAINT. Both configurations of RCA4
simulate the diurnal cycle of precipitation more accurately compared to ERAINT and
HCLIM-ALADIN. The phase of the diurnal cycle, in general, is pretty well captured over all
four subregions. In terms of precipitation intensity RCA4 underestimates rainfall from afternoon
to morning over West (Fig. 5b,c) and Central Africa (Fig. 5j,k). Reducing mixing in the
boundary layer results in flattening of the diurnal cycle over West Africa (Fig. 5b, c) while there
are almost no changes over Central Africa (Fig. 5j, k). RCA4-v1 very well simulates the diurnal
cycle over Eastern Africa with only some underestimation in early morning and afternoon (Fig.
5f). RCA4-v4 improves rainfall intensity in early morning but at the same time shows a slightly





larger underestimation in afternoon than RCA4-v1 (Fig. 5g). Over Southern Africa the RCA4
simulations at 200km are the closest to the observation (Fig. 5n,o) while the simulations at
higher resolutions underestimate the amplitude of the diurnal cycle in the afternoon.
Figure 5 clearly shows that the phase of the diurnal cycle of precipitation in Africa does not
depend on resolution but instead depends on model formulation. Both ERAINT, with the Tiedtke
convection scheme (Tiedtke, 1989), and HCLIM-ALADIN with the Bougeault scheme
(Bougeault, 1985) trigger precipitation too early during the diurnal cycle while both
configurations of RCA4 with the same Kain–Fritsch (KF) scheme (Bechtold et al., 2001)
simulate much more realistic diurnal cycle. It has previously been shown that the KF scheme is
able to reproduce late afternoon rainfall peaks for the regions where moist convection is
governed by the local forcing, for example in the southeast US (Liang, 2004) and in the tropical
South America and Africa (e.g. Bechtold et al., 2004; Da Rocha et al., 2009). Nikulin et al.,
(2012) also found that a subset of RCMs that employ the KF scheme show an improved
representation of the phase of the diurnal cycle in Africa. Our results indicate that the  impact of
resolution is only seen in the amplitude of the diurnal cycle. However, such impact is not
homogeneous across the subregions and the RCMs. For HCLIM-ALADIN, increasing resolution
lead to increasing rainfall intensity in all regions but southern Africa. RCA4 shows a similar
behaviour over West Africa, while there is a mixed response over Eastern and Central Africa.
These findings are in line with previous studies investigating resolution effects for GCMs
(Covey et al., 2016; Dirmeyer et al., 2012) and for RCMs (Walther et al., 2013). In coarser-scale
models (e.g >10km), increasing resolution only leads to changes in the magnitude, but not in the
phase of the diurnal cycle of precipitation over land.



Nevertheless, studies conducting sensitivity experiments using resolutions finer than 10 km do
find improvements in the representation of the phase (Dirmeyer et al., 2012; Sato et al., 2009;
Walther et al., 2013).

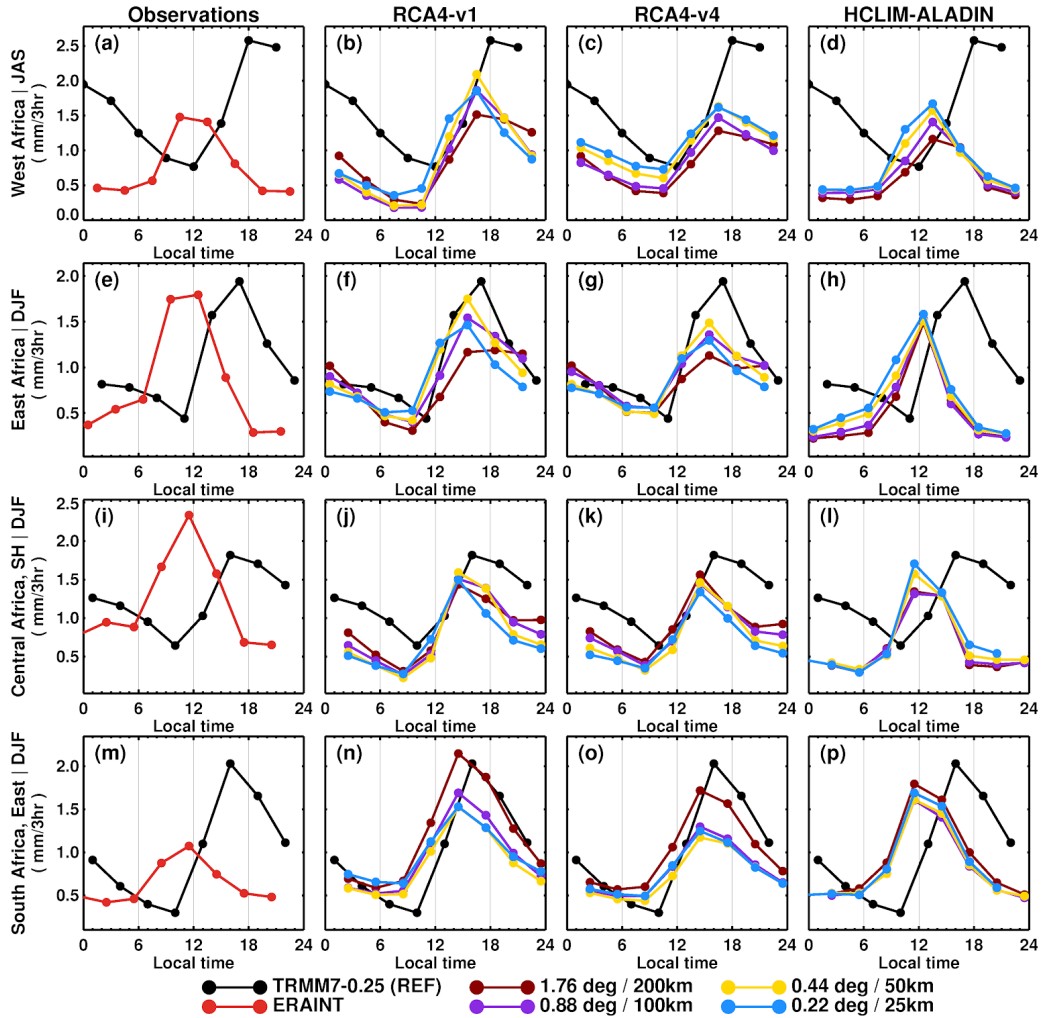

**Figure 5.** Diurnal cycle of 3-hourly mean precipitation over the four subregions in observations/ERAINT
and as simulated by RCA4 and HCLIM-ALADIN at the four different resolutions. Only land grid boxes
are used for averaging over the subregions and only wet days with more than 1mm/day are taken for
estimations of the diurnal cycle.




## 3.4 Frequency and intensity of daily precipitation

Figure 6 shows the empirical probability density function (PDF) of daily precipitation intensities
over the four subregions. The TRMM7-0.25 dataset, aggregated to the common 1.76° resolution
(TRMM7-1.76), as expected has a shorter right tail with no precipitation intensities larger than
100 mm day-1 and higher frequency for  lower intensities less than 25 mm day-1 (Fig. 6a,e,i,m).
The two TRMM7 PDFs provide reference bounds for datasets with resolution between 0.25° and
1.76°. However, uncertainties in gridded daily precipitation products in Africa are large (Sylla et
al., 2013) and we take the TRMM bounds as an observational approximation focusing more on
differences in the simulated PDFs across the four resolutions. Over West, East and central Africa
ERAINT overestimates the frequency of low (< 10 mm day-1) and extremely high (>150 mm
day-1) intensities while it underestimates the frequency of precipitation intensities in between
(Fig. 6a,e,i), especially over West Africa (Fig. 6a). In southern Africa (Fig. 6m) ERAINT
represents the frequency of daily mean precipitation more accurately compared to the other three
regions but shows almost no events with more than 150 mm day-1 in contrast to the
observations. Both RCMs, in general, have the same tendency to generate more higher-intensity
precipitation events with increasing resolution over all four subregions. In West Africa RCA4-v1
strongly underestimates the frequency of intensities with more than 20 mm day -1 at 200, 100
and 50km (Fig. 6b). A substantial improvement appears only at 25km where the right tail of the
PDF extends up to 250 mm day-1, although the frequency of precipitation events from about 50
to 150 mm day-1 is still underestimated.



The RCA4-v4 configuration markedly reduces the RCA4-v1 biases and shows more realistic
PDFs at all four resolutions (Fig. 6c). The RCA4-v4 50km simulation generates precipitation
events up to 250 mm day -1 strongly contrasting to the RCA4-v1 simulation at the same
resolution (no events more than 100 mm day-1). However, RCA4-v4 overestimates frequencies
of high intensities at 25km. Such sharp difference between two configurations of RCA4 at the
same resolution shows that model formulation also plays an important role for accurately
reproducing daily precipitation. Over West Africa all HCLIM-ALADIN simulations
overestimates the frequency of low precipitation intensities (less than 10 mm day-1) and
underestimates the frequency of intensities in the range of 10-150 mm day-1 (Fig. 6d). Similar to
RCA4, higher resolution leads to more high-intensity precipitation events in the
HCLIM-ALADIN simulations.
However, RCA4 and HCLIM-ALADIN behave in a different way with increasing resolution.
Both RCMs change the PDFs by adding more higher-intensity precipitation events extending the
right-hand tail towards higher intensities. In addition, RCA4 also increases the frequency of
medium- and high-intensity events especially going from 50 to 25km. In eastern Africa both
RCA4 configurations reproduce the observed PDFs almost perfectly (Fig. 6f, g). All four
resolutions are located within the TRMM-1.76 and TRMM-0.25 boundaries and the coarsest and
finest resolutions coincides with the respective TRMM PDFs. Contrastingly, HCLIM-ALADIN
strongly underestimates the frequency of precipitation events with more than 20 mm day-1 (Fig.
6h) over eastern Africa and even the highest 25km resolution is located below the coarse
TRMM-1.76 dataset. In central Africa both RCMs overestimate the occurrence of intensities less
than 20 mm day-1 (Fig. 6j,k,l), especially HCLIM-ALADIN (Fig. 6l) and strongly underestimate

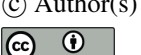



the frequency of higher-intensity events. The PDFs at all four resolutions for both RCMs are
located below the coarsest TRMM-1.76 PDF. We note that observational uncertainties in
precipitation are very large over central Africa and we should be careful in the interpretation of
Fig. 6j-l.  Seasonal mean precipitation, for example, can differ by more than 50% across different
observational datasets (Washington et al., 2013). Additionally, the TRMM dataset is scaled by
the gauge-based GPCC precipitation product while almost no long-term gauges are available in
the region (Nikulin et al., 2012).  In southern Africa RCA4 and HCLIM-ALADIN simulate the
precipitation PDFs pretty accurate (Fig. 6n-p). An interesting detail is that the 50km
HCLIM-ALADIN simulations shows higher frequency for intensities with more than 150 mm
day-1 than the 25km simulation.
In general, we see the improvement of simulated daily rainfall intensities with increasing
resolution across the African continent. There are many studies showing a similar resolution-
dependent improvement over both complex terrains and flat regions  (e.g. Chan et al., 2013;
Huang et al., 2016; Lindstedt et al., 2015; Olsson et al., 2015; Prein et al., 2016; Torma et al.,
2015a; Walther et al., 2013). Our results are in agreement with the above studies and confirm
increasing fidelity of simulated daily rainfall intensities with increasing resolution.








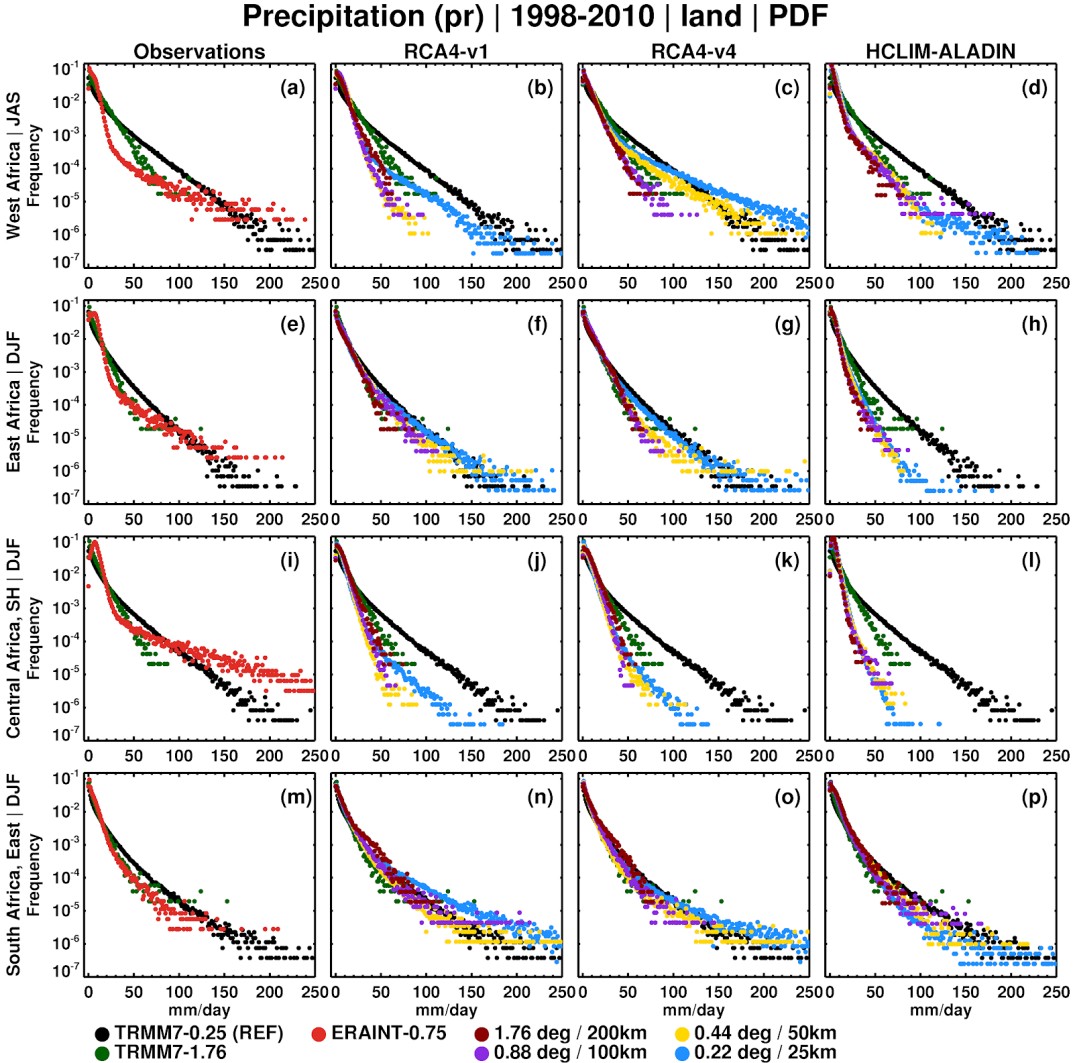

**Figure 6.** Probability distribution function of daily precipitation intensities pooled over the four subregions in observations/ERAINT and as simulated by RCA4 and HCLIM-ALADIN at the four different resolutions. TRMM7-1.76 represents TRMM7-0.25 aggregated from its native 0.25° resolution to 1.76°. A base-10 log scale is used for the frequency axis and the first bin (0-1 mm day-1) is divided by 10. Only land grid boxes are used for pooling over the subregions.






# 5 Summary and Conclusion

In this study we have investigated the impact of model formulation and spatial resolution on
simulated precipitation in Africa. A series of sensitivity, ERA-Interim reanalysis-driven
experiments, were conducted by applying two different RCMs (RCA4 and HCLIM-ALADIN) at
four resolutions (about 25, 50, 100 and 200 km). The 100km experiment, at resolution a bit
coarser than the driving ERA-Interim reanalysis, by default does not provide any
resolution-dependent added value while such added value is expected for the 50 and 25km
experiments. The 200km experiment is about 3 times upscaling of ERAINT to resolution of
many CMIP5 GCMs and should only be considered as a supplementary experiment since RCMs
do not aim to operate at such coarse resolution. In addition, to the two different RCMs, the
standard CORDEX configuration of RCA4 is supplemented by another configuration with
reduced mixing in the boundary layer. Such configuration was developed to deal with a strong
dry bias of RCA4 in Central Africa. Contrasting the two different RCMs and the two different
configurations of the same RCM at the four different resolutions allow us to separate the impact
of model formulation and resolution on simulated rainfall in Africa.
Even if the results often depend on region and season and a clear separation of the impact of
model formulation and resolution is not always straightforward, we found that model
formulation has the primary control over many aspects of the precipitation climatology in Africa.
The 100km no added value experiment shows that patterns of spatial biases in seasonal mean
precipitation are mostly defined by model formulation. These patterns are very different between
the driving ERAINT and RCMs, sometimes even with opposite sign, exemplified by the two





configurations of RCA4 in JAS (Fig. 1e-l). Resolution in general controls the magnitude of
biases and for both RCA4 and HCLIM-ALADIN higher resolution usually leads to an increase in
precipitation amount while preserving large-scale bias patterns. A side effect of such an increase
in precipitation amount is that an improvement in one region (e.g. reduction of dry biases) often
corresponds to a deterioration in another region (amplification of wet biases) as for
HCLIM-ALADIN in JAS (Fig. 1m-p). Nevertheless, on average the smallest biases in seasonal
means are found for the simulations at 50 and 25km resolution.
The impact of model formulation and resolution on the annual cycle of precipitation is mixed
and strongly depends on region and season. For example, in both West and Central Africa the
shape of the annual cycle for the 100km no added value experiment is different from ERAINT.
However, the impact of model formulation is opposite between these two regions. In West Africa
both RCMs deteriorate the ERAINT annual cycle by simulating a too early onset of the rainy
season. In contrast, over Central Africa, both models improve the ERAINT annual cycle by
reducing a strong wet bias and changing the unimodal annual cycle to a bimodal one similar to
the observations. The impact of resolution can also be different. In West and East Africa, higher
resolution (50 and 25km) leads to an improvement in the annual cycle (more realistic shape and
smaller biases). In contrast, over Central Africa, the 25km RCA4 simulations show the largest
biases while the HCLIM-ALADIN simulations at all four resolutions are almost similar.  In
general, it is difficult to conclude on a common impact of model formulation and resolution on
the annual cycle.
The phase of the diurnal cycle in Africa is completely controlled by model formulation
(convection scheme) while its amplitude is a function of resolution. Both ERAINT and



HCLIM-ALADIN shows a too early precipitation maximum around noon while RCA4 simulates
a much more realistic diurnal cycle with an evening maximum. Higher resolution does not
change the phase of the diurnal cycle but its amplitude, although the impact of resolution on the
amplitude is mixed across the four subregions and time of the day.
A pronounced and well known impact of higher resolution on daily precipitation intensities is a
more realistic distribution of daily precipitation. Our results also show that higher resolution, in
general, improves the distribution of daily precipitation. This includes reduced overestimation of
the number of days with low precipitation intensities and reduced underestimation of the number
of days with high intensities. The latter results in extending the right-hand tail of the distribution
towards higher intensities similar to observations. This also means that at higher resolutions the
time mean climate (e.g. seasonal mean and annual cycle) is made up of more realistic
underpinning daily precipitation than at lower resolutions. It is also worth emphasizing that if
low resolution models are not able to simulate high rainfall days then it will be difficult for them
to say anything robust about projected climate changes in high rainfall events. However,
regionally, model formulation can also play an important role in the distribution of daily
precipitation. For example, in West Africa the 50km RCA4-v4 configuration with reduced
mixing in the boundary layer shows a remarkable improvement in the shape of the PDF (Fig. 1c)
compared to the standard RCA4-v1 configuration at the same resolution (Fig 1b). Moreover, the
RCA4-v4 configuration at 50 km shows almost the same PDF as RCA4-v1 at 25km. Such
contrast indicates that for daily precipitation intensities model formulation can have the same
impact as doubled resolution.



Improvements in simulated precipitation in high resolution RCMs relative to coarse-scale GCMs
are often attributed as being an resolution-dependent added value of downscaling. Our results
show that for Africa improvements are not always related to higher resolution but also to
different model formulation between the RCMs and their driving reanalysis. A common
framework for quantifying added value of downscaling is to evaluate some aspect of the climate
in high-resolution RCM simulations and in their coarse-resolution driving reanalysis or GCMs
over a historical period (Di Luca et al., 2015; e.g. Hong and Kanamitsu, 2014; Rummukainen,
2016). If the RCM simulations show smaller biases compared to reference observations than the
driving GCMs, one can conclude that RCMs provide an added value and vice versa. However,
such a framework does not separate the impact of different model formulation between RCMs
and their driving GCMs and higher resolution in the RCM simulations. Our results indicate that
improvements in RCM simulations may simply be related to different model formulation and not
necessarily to higher resolution.  In general, model formulation related improvements cannot be
considered as an added value of downscaling as such improvements are strongly model
dependent and cannot be generalised.
Within commonly used RCM evaluation framework, e.g. the CORDEX evaluation experiment,
it is not straightforward, if possible at all, to isolate the impact of model formulation and
resolution in RCM simulations. We show that running RCMs at about the same resolution as a
driving reanalysis (e.g. ERAINT at about 80km or ERA5 at about 30km) helps to separate the
impacts of model formulation and higher resolution in dynamical downscaling. We propose that
such a simple additional experiment can be an integral part of the RCM evaluation framework in
order to elucidate the added value of downscaling.




# Code availability

The analysis is done in MATLAB and IDL and codes can be provided by request as they are but
without support on implementing them in another computing environment.

# Data availability

The ERA-Interim reanalysis is available at https://apps.ecmwf.int/datasets/, the GPCC dataset is
available at https://www.dwd.de/EN/ourservices/gpcc/gpcc.html, the CRU dataset is available at
https://catalogue.ceda.ac.uk/uuid/5dca9487dc614711a3a933e44a933ad3 , the UDEL dataset is
available at http://climate.geog.udel.edu/~climate/html_pages/download.html, the TRMM
dataset is available at https://pmm.nasa.gov/data-access/downloads/trmm. The RCA4 and
HCLIM-ALADIN data can be provided by request.

# Author contribution

MW performed RCA4 simulations and all the analysis and wrote the initial draft. GN developed
the experiment design and provided guidance for the analysis. EK and GN revised the initial
draft. CJ is responsible for setting up the new RCA4 configuration (v4). DB and DL are
responsible for performing the HCLIM-ALADIN simulations over Africa. All the authors
contributed with discussions and revisions.

# Conflict of interest

There is no conflict of interest in this study.

# Acknowledgements

This work was done with support from the FRACTAL (www.fractal.org.za) and AfriCultuReS
(http://africultures.eu/) projects. FRACTAL is part of the multi-consortia Future Climate for
Africa (FCFA) programme - jointly funded by the UK's Department for International
Development (DFID) and the Natural Environment Research Council (NERC). AfriCultuReS
has received funding from the European Union's Horizon 2020 research and innovation
programme under grant agreement No 774652. The authors thank the European Centre for
Medium-Range Weather Forecasts (ECMWF), the Global Precipitation Climatology Centre
(GPCC), the British Atmospheric Data Centre (BADC), the University of East Anglia (UEA),
the University of Delaware and the Goddard Space Flight Center (GSFC) for providing data. All





simulations were conducted on the supercomputer in the National Supercomputer Centre,
Linköping, Sweden.







































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
