# Peer review of "The impact of regional climate model formulation and"

_Earth System Dynamics, 2019_

## Referee Comment (RC1) · Anonymous Referee #1 · 12 Nov 2019

12/11/2019

Manuscript ID: ESD-2019-55

Title: The impact of RCM formulation and resolution on simulated precipitation in Africa

Author(s): Minchao Wu et al.

Recommendation: Accepted with major revisions

General Comments: This paper investigates the impacts of the model formulation and resolution on the ability of two Swedish RCMs to simulate precipitation in Africa. The two RCMs were used at 200, 100, 50 and 25 km resolutions and one of them has two different formulations. This experimental setup allows disentangling the improvements

related to either the resolution or the model formulation. The topic is of interest and relevant for the RCM community and deserve to be considered for publication. However, I am not sure that the journal Earth System Dynamics is the best journal to convey this study since I very rarely read RCM papers from that journal. I let the editor to decide whether the topic of this paper is suitable for this journal or not.

The paper is very well written and the literature review is very good although the introduction could include more papers related to the topic. Few papers suggested below could be added in the literature review of the introduction. The abstract is generally fine, but few sentences are not clear and should be improved. The introduction is generally clear and interesting, but it should be improved to emphasize the full motivation of the analysis. The methodology is appropriate to address the objectives of the study, but I am concern about the relevancy to run an RCM at 200 and 100 km and the utility of those simulations in the paper. The results are interesting and address the objectives raised at the beginning of the paper. The figures are clear and support the analysis. The conclusions are in line with the analysis and are of interest for the community. Thus, I recommend this paper to be accepted with major revisions.

Major Comments: 1. Introduction: The introduction is interesting and fully explain the motivation of the study. However, it is a bit short and it lacks a more complete literature review of the challenges to simulate precipitation over Africa. Thus, I recommend to extend the paragraph from the line 98 to 117 in 2 or 3 paragraphs to include more RCM studies that paid attention to the challenges to simulate precipitation in Africa with RCMs.

Here is a short list giving examples of papers that could be added to the literature review in the introduction: Evaluation of present-day rainfall simulations over West Africa in CORDEX regional climate models, by Akinsanola, A.A and Ogunjobi, K.O.

Spatial distribution of precipitation annual cycles over South Africa in 10 CORDEX regional climate model present-day simulations, by Favre, A.et al.

Assessing the Capabilities of Three Regional Climate Models over CORDEX Africa in Simulating West African Summer Monsoon Precipitation, by Akinsanola, A.A. et al.

Simulation of the West African monsoon onset using the HadGEM3-RA regional climate model, by Diallo, I. et al.

Improving the simulation of the West African monsoon using the MIT regional climate model, by Im, E.-S., Gianotti, R.L., Eltahir, E.A.B.

Assessment of the performance of CORDEX regional climate models in simulating East African rainfall, by Endris, H.S.et al.

Downscaling reanalysis over continental Africa with a regional model: NCEP versus ERA Interim forcing, by Druyan, L.M, Fulakeza, M.

Climate simulation over CORDEX Africa domain using the fifth-generation Canadian Regional Climate Model (CRCM5), by Hernández-Díaz, L. et al.

Evaluation of rainfall simulations over Uganda in CORDEX regional climate models, by Kisembe, J. et al.

The diurnal cycle of precipitation in regional spectral model simulations over West Africa: Sensitivities to resolution and cumulus schemes, by He, X. et al.

2. Methodology: Even with the warnings at lines 183-184 and 488-490, I really wonder if it is relevant to use an RCM at 200 and 100 km resolutions and I also wonder if the use of those simulations adds substantial information to the paper. I think that 200 and 100 km are excessively far from the RCM range of resolution or comfort zone for which it is configured and calibrated and I think that little is gained from those simulations in this paper. Thus, either the authors should be really convincing that those resolutions are relevant and add substantial content to the paper or either they should remove at least the 200 km resolution simulations from the paper. In some sense, the 100-km resolution simulations may be relevant since they are at a resolution close to ERA-Interim and can be used as a "no added-value experiment". Additionally, by removing

the 200 km resolution simulation, only an aggregation to 100 km would be necessary for the analysis, leading to more details of the simulated precipitation in the results section.

3. Due to the large African domain and that no spectral nudging was used, I am wondering if the internal variability as mentioned in the line 175 would be large enough to produce large differences between simulations from the same model ? Thus, I would suggest the authors to rerun the 50-km resolution simulation of one of the two RCMs with different initial conditions or different starting time and repeat the analysis to see if the IV could affect the simulated precipitation.

4. Conclusion: The discussion of the results in the conclusion is a bit thin and the opening towards additional studies that could follow that one is missing. I would suggest the authors to add some discussions about the results and provide few ideas towards additional studies that could follow that one.

Minor Comments: 1. Title: I think that abbreviations should not be used in titles in general. Thus, I suggest to replace "RCM" by "regional climate model" in the title. 2. Lines 25 and 183: Please add a "–" after ALADIN or use parentheses to name the two models. 3. Line 27-29 and 42-43: Something is wrong with these sentences. Please correct them. 4. Line 32-34 and 35-39: The sentences are not clear and some points are repeated. Please improve all the sentences of those lines and simplify the message conveyed. 5. Lines 66, 72, 88, 94, 97, 122, 173 etc: The word "results" is used too many times, is too vague and sometimes inappropriate. Sometimes it means the outcome of downscaling. In another context, it refers to the outcome of the analysis. I would suggest to use other words to avoid confusion. As instance, the word "simulation" could be used at lines 66, 72, 173. Please pay attention to every time the word "results" is used and consider using another word or changing the sentence to be more specific. 6. Line 92 and 464. Torma et al. 2015 a and b are the same paper. In addition, I believe that the paper of Giorgi et al. (2016) is more appropriate giving the context. Giorgi, F., Torma, C., Coppola, E., Ban, N.,

Schar, C., and Somot,S.: Enhanced summer convective rainfall at Alpine high eleva-tions in response to climate warming, Nat. Geosci., 9, 584–589,2016. 7. Lines 113 and 143: Remove "e.g." 8. Line 146 and the rest of the paper: About the use of RCA4-v1 and RCA4-v4 to distinguish the two RCA model formulations. I think that v4 is not the best way to name the reduced turbulent mixing simulation since 4 brings in mind that a v2 and v3 are existing and that they are not used in this paper. I would suggest to use RCA4 and RCA4-RTB for Reduced Turbulent Mixing to name the two RCA simulations. 9. Lines 171-178: I am confuse here about the size of the domains at different resolutions. Is the size of the free domain or full domain including the nudging zone the same between the simulations? Moreover, at line 175, it is mentioned that an additional experiment at $0.88°$ was performed, but this experiment is never mentioned later on in the analysis. Maybe the sentence of the line 176-178 refer to the two $0.88°$ simulations. Please pay attention to all the sentences of those lines and specify clearly, which simulation are referred. 10. Line 180-181: For these simulations . . .. Please specify which simulations? 11. Table 1. What the small "a" after 222x222 means? 12. Line 203: Please specify the time period covered by TRMM and be more specific on the time period used for the analysis of Figures 5-6. I think that TRMM starts in 1997 or 1998. Moreover, considering the little amount of weather stations in Africa that are used to create CRU, UDEL and GPCC, I think that TRMM figures covering a subset of the full 1981-2010 could be used in Figures 2, 3 and 4 as it is done in Nikulin et al. (2012). 13. Line 229: Replace "most northern" by "northernmost". 14. Lines 237-239: Please improve the sentence that is not clear. 15. Figure 2 and 3: Color scale on the left: The values above 15 mm/day could be removed as in Nikulin et al. (2012) 16. Figure 2 and 3: Color scale at the bottom: I would suggest to use a white color between -0.5 and 0.5. This would prevent the color change at 0 that is misleading. As example, the Sahara desert is sometimes yellow or blue because there is almost no precipitation falling there. 17. Figure 2 caption: Please emphasize that the values are aggregated at 200 km. 18. Line 284-286: Please give more details about the statement here. 19. Lines 333-335: Please clarify what is meant by "completely opposite behavior". 20.

Lines 458-460: It is not clear to me that the 50 km HCLIM simulation shows higher frequency than the 25 km HCLIM simulation. 21. Figure 6: Please emphasize in the caption that the season is different for the different regions. 22. Lines 540-541: There are mistakes about the Figure numbers. 23. Reference: Please remove the capital letters of the title of Sylla et al. (2013).

―――――――――――――――――――

---

## Referee Comment (RC2) · John Scinocca (Referee) · 16 Dec 2019

In this study the authors introduce a procedure to separate the impact of model formulation from the impact of resolution on the dynamical downscaling results of regional climate models (RCMs) driven by observations (reanalyses). The procedure involves performing the downscaling at several horizontal resolutions. The coarsest RCM resolution is set to match the resolution of the reanalysis model that provides the driving data. This is referred to as the "no added-value experiment", which I will refer to as the NAVE. The authors make the point that the NAVE biases vs the reanalysis biases (relative to an independent observational dataset) result from "model formulation" differences and so are independent of added value. Once NAVE biases are defined, higher resolution RCM simulations are employed to document the evolution of NAVE

biases with resolution. It is argued that a reduction of NAVE biases with increasing resolution indicates added value in the RCM. The authors employ this procedure to precipitation biases in RCM downscaling results over the African CORDEX domain from two regional models.

The results of the authors' analysis of model formulation vs resolution is often mixed with few clear results. But this is overshadowed by the introduction of the NAVE procedure itself, which is highly publishable as it provides a tool to the RCM community to make progress on the complex issue of added value in RCM studies. In fact, the NAVE approach would seem to have a logical extension to the much more important issue of value added by RCMs in climate-change experiments. In my detailed comments, I suggest a generalization of the NAVE approach to the issue of value added by RCMs in climate-change experiments. It is my recommendation that this manuscript be accepted for publication with only minor revision.

General Minor Comments

1) NAVE procedure applied to Climate-Change experiments:

The NAVE procedure would seem to be equally applicable to climate change problems to help distinguish the impact of model formulation from the impact of resolution on RCM climate-change responses relative to those of its driving GCM. In the climate-change context, two sets of RCM runs would need to be performed - NAVE runs at the resolution of the driving global climate model (GCM) and the usual high-resolution runs used for downscaling GCM climate-change results. Consider a typical time-slice experiment over a CORDEX domain performed at the end of the 20th and 21st centuries. For a given climate index (eg screen-level temperature, precipitation, extremes etc.), one could construct the three climate-change responses:

$R\_GCM(X) = GCM\_21st(X) - GCM\_20th(X)$

$R\_NAVE(X) = NAVE\_21st(X) - NAVE\_20th(X)$

R_RCM(X) = RCM_21st(X) - RCM_20th(X),

where each term on the right is a time (and/or ensemble) average at a given spatial location "X".

In the above, R_NAVE(X) represents the climate-change signal associated with model formulation differences between the RCM and GCM. As for the authors' present-day analysis, the potential for value added due to the response associated with resolution changes may be expressed as:

R_RES(X) = R_RCM(X) - R_NAVE(X).

The NAVE analysis allows the decomposition:

R_RCM = R_NAVE(X) + R_RES(X)

Given R_RES(X), and R_NAVE one can ask interesting questions like:

- Where is R_RES(X) significant in the RCM domain?

- Do these location correlate well with where the authors found downscaling improvement in their NAVE analysis of reanalysis driven RCMs?

- Where is R_RCM appreciably different from R_NAVE? The appreciable difference analysis presented in Section 5 of Scinocca et al. 2015 (JClim p. 17-35) would seem like an ideal approach to address this question. In locations where there exists an appreciable difference, there exists the potential for added value. However, where there is no appreciable difference, there can be no added value - irrespective of how one chooses to define added value.

This is in line with the authors' stated goals (ll.116-118). Clearly such climate-change questions are outside the authors' present study but, they may want to discuss this potential application of the NAVE approach for future investigation.

2) Interpretation of the NAVE:

[Figure]

It is assumed here that differences in the NAVE and driving model results arise from differences the RCM and GCM model formulation. This would be strictly true only if the RCM were also run in a global mode. The one-way nesting approach introduces a number of potential artifacts which are most acute for large RCM domains and applications that do not use interior (or spectral) nudging - both of which are the case for the authors' present study (eg Section 2 of Scinocca et al. 2015 JClim p. 17-35). The authors should acknowledge this issue when introducing the NAVE.

3) RCM model tuning:

ll.183-185 "We note that in general, both regional models - RCA and HCLIM-ALADIN were developed to operate at a range of 10-50km resolution and their performance at 100 and 200km may not be optimal." This is a non-trivial point, given the philosophy of the authors' NAVE approach. Where there is systematic improvement of NAVE biases with increased resolution, the authors interpret this as a systematic increase in added value. However, The poorer results at the coarser resolutions may also be related to a lack of model retuning at these non-standard resolutions. Very few physical parameterizations are automatically scale dependent and an adjustment of their free parameters with changing spatial resolution should in principle be performed.

Retuning the RCMs at each spatial resolution would represent a significant undertaking and these added degrees of freedom would complicate the main point made in this study. Consequently, I would recommend that this issue be addressed by simply having it raised as a caveat.

4) Interior nudging:

In downscaling reanalysis products, the authors chose not to employ any constraints on the interior RCM solution such as spectral nudging (ll.185-186). In focusing on such evaluation experiments, one could argue that it is more appropriate to use spectral nudging to constrain the large scales to obtain the best downscaled results in their study. Any upscale influence produced by the RCM would serve only to degrade the

large scale flow as it is well observed and represented in reanalysis produces. By not constraining the RCM in this way, the authors leave open the possibility that locations of large biases in their high-resolution RCM results are due to the downscaling of the wrong large-scale flow rather than a lack of intrinsic added value. For more detail see Section 2 of Scinocca et al. 2015 (JClim p. 7-35).

Detailed Minor Comments

l.26 "Additionally to the two RCMs" perhaps change to "In addition to the two RCMs"

l.31 "the phase of the diurnal cycle is" perhaps change to "the phase of the diurnal cycle in precipitation is"

l.71 "However, added value from RCMs" should be changed to "However, perceived added value from RCMs" for the context of the sentence.

ll.141-147. It was unclear whether the difference between v1 and v4 was simply a change in a free parameter for an existing scheme or whether the difference was associated with a change in the equations of the scheme. The former might be considered "tuning" while the latter considered a "formulation" difference.

ll.176-178 It would be helpful to show these plots to see if the differences have any correlation with later results (perhaps in an appendix) - particularly the distribution of temperature differences.

ll.260-262 Fig 2b-p. It was often hard to associate the location of a particular bias with the full field in panel a. Expressing the bias as a percentage difference from the full field would be helpful in the West and central regions. However, where there is weak precipitation in the reference/obs data this may be problematic.

ll.350-352 Fig 4. It would be better to use the colour red for the reference GPCC7 curves in this figure. I had difficulty seeing the GPCC7 curves in a number of the model result panels in columns 2-4.

ll.558-560 "In general, model formulation related improvements cannot be considered as an added value of downscaling as such improvements are strongly model dependent and cannot be generalised." Also, such formulations could in principle be used in global models and so obviate the need for the RCM.
* * *

---

## Author Comment (AC1) · 16 Feb 2020

We would like to thank both reviewers for their helpful and useful comments on the manuscript "The impact of RCM formulation and resolution on simulated precipitation in Africa" by Wu et al. (esd-2019-55).

Below is our response to the reviewers' comments, following the structure: (1) comments from Referees (2) author's response, (3) author's changes in manuscript are in ""

Comments from anonymous Referee #1

Recommendation: Accepted with major revisions General Comments:

[Figure]

This paper investigates the impacts of the model formulation and resolution on the ability of two Swedish RCMs to simulate precipitation in Africa. The two RCMs were used at 200, 100, 50 and 25 km resolutions and one of them has two different formulations. This experimental setup allows disentangling the improvements related to either the resolution or the model formulation. The topic is of interest and relevant for the RCM community and deserve to be considered for publication. However, I am not sure that the journal Earth System Dynamics is the best journal to convey this study since I very rarely read RCM papers from that journal. I let the editor to decide whether the topic of this paper is suitable for this journal or not. The paper is very well written and the literature review is very good although the introduction could include more papers related to the topic. Few papers suggested below could be added in the literature review of the introduction. The abstract is generally fine, but few sentences are not clear and should be improved. The introduction is generally clear and interesting, but it should be improved to emphasize the full motivation of the analysis. The methodology is appropriate to address the objectives of the study, but I am concern about the relevancy to run an RCM at 200 and 100 km and the utility of those simulations in the paper. The results are interesting and address the objectives raised at the beginning of the paper. The figures are clear and support the analysis. The conclusions are in line with the analysis and are of interest for the community. Thus, I recommend this paper to be accepted with major revisions.

Major Comments:

1. Introduction: The introduction is interesting and fully explain the motivation of the study. However, it is a bit short and it lacks a more complete literature review of the challenges to simulate precipitation over Africa. Thus, I recommend to extend the paragraph from the line 98 to 117 in 2 or 3 paragraphs to include more RCM studies that paid attention to the challenges to simulate precipitation in Africa with RCMs.

Here is a short list giving examples of papers that could be added to the literature review in the introduction: ...

[Figure]

Response: We completely agree with this comment and extended Introduction by including more RCM studies. There are really many RCM-based evaluation studies for Africa and we focus mostly on studies with large RCM ensembles.

2. Methodology: Even with the warnings at lines 183-184 and 488-490, I really wonder if it is relevant to use an RCM at 200 and 100 km resolutions and I also wonder if the use of those simulations adds substantial information to the paper. I think that 200 and 100 km are excessively far from the RCM range of resolution or comfort zone for which it is configured and calibrated and I think that little is gained from those simulations in this paper. Thus, either the authors should be really convincing that those resolutions are relevant and add substantial content to the paper or either they should remove at least the 200 km resolution simulations from the paper. In some sense, the 100-km resolution simulations may be relevant since they are at a resolution close to ERAInterim and can be used as a "no added-value experiment". Additionally, by removing the 200 km resolution simulation, only an aggregation to 100 km would be necessary for the analysis, leading to more details of the simulated precipitation in the results section.

Response: From the beginning, our experiment was developed to include simulations at coarse resolution outside of a RCM comfortable zone following experiment setup in Moufouma-Okia et al. (2015) with the coarsest resolution - 150km for their RCM (HadGEM3-RA). Our point of view is that such coarse-resolution simulations are a useful supplement to simulations at RCM comfortable resolution and help us to understand RCM behaviour without additional, resolution-dependent tuning. Our results show that performance of the RCA4 and ALADIN RCMs at 200km is, in general, consistent and fits well to what can be expected for moving from the highest (25km) to coarsest (200km) resolution. This, for example, includes among others i) a common tendency to precipitate less at coarser resolution for both RCA4 and ALADIN in JAS (Fig. 2) and ii) the deterioration of simulated daily rainfall intensities with decreasing resolution (Fig. 6). We also found different behaviour of RCA4 and ALADIN with decreasing resolution in DJF (Fig. 3). This shows that the impact of coarser resolution on

the simulated precipitation climatology is not the same in different seasons and regions and depends on RCM formulation. We would prefer to keep the coarse resolution simulations as the study becomes less complete if the 200km simulations are excluded.

We also need to note that running a RCM at resolution outside of its comfortable resolution range can sometimes bring unexpected results, different from what was previously thought. Vergara-Temprado et al. (2020) found that an explicit representation of convection in a RCM (CCLM) may be beneficial in representing some aspects of climate over Europe at 12-25km resolution that is far away from a few km resolution typical for convection permitting RCMs. Running a hydrostatic RCM (RCA3) in the grey zone (6.5 km), Güttler et al. (2015) showed that many aspects of precipitation climatology over Europe are improved at 6.5km resolution compared to coarser resolutions (50, 25 and 12,5km). Vergara-Temprado et al. (2020) https://journals.ametsoc.org/doi/pdf/10.1175/JCLI-D-19-0286.1 Güttler et al. (2015) https://journals.ametsoc.org/doi/full/10.1175/MWR-D-14-00302.1

We added a short explanation to "2.2 Experiment design":

"We note that in general, both regional models - RCA and HCLIM-ALADIN were developed to operate at a range of tens of km resolution and their performance at 100 and especially at 200km may not be optimal. A potential caveat here is that very few RCM physical parameterisations are automatically scaled at very coarse resolution. Thus, results at the coarser resolutions may be partly related to the lack of model retuning. We think that such coarse-resolution simulations are a useful supplement to simulations at a RCM comfortable resolution zone and help us to understand RCM behaviour without additional, resolution-dependent tuning. "

3. Due to the large African domain and that no spectral nudging was used, I am wondering if the internal variability as mentioned in the line 175 would be large enough to produce large differences between simulations from the same model ? Thus, I would suggest the authors to rerun the 50-km resolution simulation of one of the two RCMs

with different initial conditions or different starting time and repeat the analysis to see if the IV could affect the simulated precipitation.

Response: To respond to this and to other experiment-related comments from both reviewers we extended section "2.2 Experiment design" by providing more details on our experiment setup, potential caveats and additional sensitivity experiments. We also performed two additional simulations (RCA4-v1 and ALADIN) at 50km resolution but starting them on 1st January 1980 instead of 1st January 1979 as for all other simulations in the study (see updated "2.2 Experiment design" ).

4. Conclusion: The discussion of the results in the conclusion is a bit thin and the opening towards additional studies that could follow that one is missing. I would suggest the authors to add some discussions about the results and provide few ideas towards additional studies that could follow that one.

Response: We agree with this comment. In context of more discussions, we should note that there are almost no studies focusing on multi-resolution RCM experiments over Africa, including an analog of the no added value experiment (NAVE). We've already proposed that the NAVE can be used as an additional experiment within the CORDEX framework. In the revised manuscript we also added that the next step is to focus on i) other variables and especially on processes and ii) on applications of the NAVE for RCM-based future climate projections (many thanks to John Scinocca for providing an detailed description of the NAVE in the climate projection context).

"In our study, as the first step, we focus only on precipitation that has large relevance for climate change impact studies. As the next step, we foresee similar studies looking also at other variables and especially at processes and drivers relevant for regional climate.

Moreover, the same NAVE framework can be used for quantifying the added value in RCM-based future climate projections. For this, one needs to downscale GCMs at their native resolution in addition to the standard CORDEX resolutions (25 or 50km).

The RCM projections at the native GCM resolution serve as the NAVE in the climate change context. A potential caveat, already mentioned in our study, is that RCMs are generally developed and tuned to operate at resolution of tens of km. "Downscaling" a GCM at its native resolution, for example 150 or 200km, may lead to artefacts related to a lack of RCM retuning for coarser resolution. Nerveless, more and more GCMs, for example in CMIP6, have resolution finer than 100km that allows application of the NAVE. "

Minor Comments: 1. Title: I think that abbreviations should not be used in titles in general. Thus, I suggest to replace "RCM" by "regional climate model" in the title.

Response: We changed the title accordingly.

2. Lines 25 and 183: Please add a "–" after ALADIN or use parentheses to name the two models.

Response: changed to (SMHI-RCA4 and HCLIM38-ALADIN)

3. Line 27-29 and 42-43: Something is wrong with these sentences. Please correct them.

Response: We reformulated these sentences:

l. 27-29 is now "By contrasting different downscaling experiments, it is found that model formulation has the primary control over many aspects of the precipitation climatology in Africa."

l. 42-43 is now "Such model formulation related improvements are strongly model dependent and can, in general, not be considered as an added value of downscaling."

4. Line 32-34 and 35-39: The sentences are not clear and some points are repeated. Please improve all the sentences of those lines and simplify the message conveyed.

Response: We reformulated these sentences:

"However, the impact of higher resolution on the time mean climate is mixed. An improvement in one region/season (e.g. reduction of dry biases) often corresponds to a deterioration in another region/season (e.g. amplification of wet biases). At the same time, higher resolution leads to a more realistic distribution of daily precipitation. Consequently, even if the time-mean climate is not always greatly sensitive to resolution, the realism of the simulated precipitation increases as resolution increases."

5. Lines 66, 72, 88, 94, 97, 122, 173 etc: The word "results" is used too many times, is too vague and sometimes inappropriate. Sometimes it means the outcome of downscaling. In another context, it refers to the outcome of the analysis. I would suggest to use other words to avoid confusion. As instance, the word "simulation" could be used at lines 66, 72, 173. Please pay attention to every time the word "results" is used and consider using another word or changing the sentence to be more specific.

Response: This comment has been taken into account. We made a number of changes and tried to use "results" for describing the outcome of the analysis.

6. Line 92 and 464. Torma et al. 2015 a and b are the same paper. In addition, I believe that the paper of Giorgi et al. (2016) is more appropriate giving the context. Giorgi, F., Torma, C., Coppola, E., Ban, N.,Schar, C., and Somot,S.: Enhanced summer convective rainfall at Alpine high elevations in response to climate warming, Nat. Geosci., 9, 584–589,2016.

Response: We think that both studies are relevant in the given context and added Giorgi et al. (2016) as well.

7. Lines 113 and 143: Remove "e.g."

Response: removed

8. Line 146 and the rest of the paper: About the use of RCA4-v1 and RCA4-v4 to distinguish the two RCA model formulations. I think that v4 is not the best way to name the reduced turbulent mixing simulation since 4 brings in mind that a v2 and v3 are

existing and that they are not used in this paper. I would suggest to use RCA4 and RCA4-RTB for Reduced Turbulent Mixing to name the two RCA simulations.

Response: At the moment there are three RCA4 configurations (small domain-related retuning) used in CORDEX and available through ESGF, namely: v1, v2 and v3. RCA4-v4 is a new configuration developed to deal with a large dry bias in Central Africa. New simulations generated by the RCA4-v4 for the Africa-CORDEX domain will be also available on ESGF and we would prefer to keep RCA-v1 and RCA-v4 for consistency. We also added necessary explanations.

"RCA4 has three configurations used for CORDEX simulations that are available through ESGF. They are named (so called RCM version) as v1 (Europe, Arctic, Africa, Southeast Asia, Central and North America), v2 (South Asia) and v3 (South America) and differ in some domain-specific re-tuning. In this study we also include a new configuration - v4. The RCA-v4 is based on RCA4-v1 but with a change in one parameter leading to reduced turbulent mixing in stable situations (especially momentum mixing). Such change in the parameter was applied to reduce a prominent dry bias found in the RCA4-v1 CORDEX Africa simulations over Central Africa (Tamoffo et al., 2019; Wu et al., 2016). Using two parameter settings of RCA4 allows us to examine how sensitive our results are to such small tuning of the same RCM."

9. Lines 171-178: I am confuse here about the size of the domains at different resolutions. Is the size of the free domain or full domain including the nudging zone the same between the simulations? Moreover, at line 175, it is mentioned that an additional experiment at 0.88âŮę was performed, but this experiment is never mentioned later on in the analysis. Maybe the sentence of the line 176-178 refer to the two 0.88âŮę simulations. Please pay attention to all the sentences of those lines and specify clearly, which simulation are referred.

Response: Table 1 shows the size of the full domain including the 8 grid point relaxation zone in all directions that is actually explained in l. 169-173. We also updated the title

for Table 1. "Table 1. The full domain configuration and time step for the RCA4 and HCLIM-ALADIN simulations. The full domain includes the 8 grid point relaxation zone."

l. 175: We extended 2.2 Experimental design adding necessary explanations.

10. Line 180-181: For these simulations . . ... Please specify which simulations?

Response: changed to "for these NAVE simulations"

11. Table 1. What the small "a" after 222x222 means?

Response: It's a typo, deleted.

12. Line 203: Please specify the time period covered by TRMM and be more specific on the time period used for the analysis of Figures 5-6. I think that TRMM starts in 1997 or 1998. Moreover, considering the little amount of weather stations in Africa that are used to create CRU, UDEL and GPCC, I think that TRMM figures covering a subset of the full 1981-2010 could be used in Figures 2, 3 and 4 as it is done in Nikulin et al. (2012).

Response: The TRMM 3B42 (v7) precipitation dataset provides satellite-based precipitation estimates adjusted by large-scale monthly precipitation from gauge networks that is, in our case, the GPCC product. This means that monthly mean TRMM and GPCC7 precipitation in general do not differ too much and are basically almost the same if remapped to the same resolution or averaged over a region. The TRMM data set is used in the study only because of availability of daily and 3-hr precipitation for evaluation of the simulated daily and 3-hr precipitation but not to evaluate seasonal means and annual cycle. We added a few lines to 2.3 "Observations and reanalysis" to explain this issue. "The TRMM product starts in 1998 and for evaluation of precipitation extremes and diurnal cycle we use a shorter period (1998-2010) in contrast to 1981-2010 used for evaluation of seasonal means and annual cycle. We also need to note that the TRMM 3B42-v7 precipitation product provides satellite-based precipitation estimates adjusted by the GPCC gauge-based precipitation. This means that

monthly mean TRMM 3B42 and GPCC precipitation are almost the same if remapped to the same resolution or averaged over a region."

13. Line 229: Replace "most northern" by "northernmost".

Response: replaced

14. Lines 237-239: Please improve the sentence that is not clear.

Response: reformulated

"RCA4-v4 shows a similar pattern compared to RCA4-v1 but substantially reduces the dry bias over Central Africa at all four resolutions (Fig. 2i-l). For both configurations of RCA4, the smallest dry bias is found at the highest 25km resolution. At the same time, an overestimation of precipitation north of the region with the dry bias becomes more pronounced, especially for RCA4-v4."

15. Figure 2 and 3: Color scale on the left: The values above 15 mm/day could be removed as in Nikulin et al. (2012)

Response: We limited the color scale by 18 mm/day as there are a few grid boxes with values slightly above 17 mm/day.

16. Figure 2 and 3: Color scale at the bottom: I would suggest to use a white color between -0.5 and 0.5. This would prevent the color change at 0 that is misleading. As example, the Sahara desert is sometimes yellow or blue because there is almost no precipitation falling there.

Response: We agree and use white color between -0.5 and 0.5 mm/day.

17. Figure 2 caption: Please emphasize that the values are aggregated at 200 km.

Response: added

18. Line 284-286: Please give more details about the statement here.

Response: We added a sentence after.

"This indicates that HCLIM-ALADIN parameterisations may be better suited to work also at coarser resolution."

19. Lines 333-335: Please clarify what is meant by "completely opposite behavior".

Response: These lines were reformulated:

"HCLIM-ALADIN maintains similar behavior to that in Eastern Africa, although the difference in precipitation across the resolutions is small (Fig. 4l). On the other hand, for both configurations of RCA4 in Central Africa, increasing resolution leads to decreasing precipitation during the rainy seasons, especially in January."

20. Lines 458-460: It is not clear to me that the 50 km HCLIM simulation shows higher frequency than the 25 km HCLIM simulation.

Response: We checked once more and saw the same result: the 50km PDF (yellow) is above the 25km one (blue) and even for a wider range of intensities (50 to about 200 mm/day) than we noted first. We changed these lines accordingly:

"An interesting detail is that the 50km HCLIM-ALADIN simulation shows higher frequency for intensities in the range of 50 to about 200 mm/day than the 25km simulation."

21. Figure 6: Please emphasize in the caption that the season is different for the different regions.

Response: added

22. Lines 540-541: There are mistakes about the Figure numbers.

Response: fixed

23. Reference: Please remove the capital letters of the title of Sylla et al. (2013).

Response: removed

Comments from John Scinocca

Major Comments:

In this study the authors introduce a procedure to separate the impact of model formulation from the impact of resolution on the dynamical downscaling results of regional climate models (RCMs) driven by observations (reanalyses). The procedure involves performing the downscaling at several horizontal resolutions. The coarsest RCM resolution is set to match the resolution of the reanalysis model that provides the driving data. This is referred to as the "no added-value experiment", which I will refer to as the NAVE. The authors make the point that the NAVE biases vs the reanalysis biases (relative to an independent observational dataset) result from "model formulation" differences and so are independent of added value. Once NAVE biases are defined, higher resolution RCM simulations are employed to document the evolution of NAVE biases with resolution. It is argued that a reduction of NAVE biases with increasing resolution indicates added value in the RCM. The authors employ this procedure to precipitation biases in RCM downscaling results over the African CORDEX domain from two regional models. The results of the authors' analysis of model formulation vs resolution is often mixed with few clear results. But this is overshadowed by the introduction of the NAVE procedure itself, which is highly publishable as it provides a tool to the RCM community to make progress on the complex issue of added value in RCM studies. In fact, the NAVE approach would seem to have a logical extension to the much more important issue of value added by RCMs in climate-change experiments. In my detailed comments, I suggest a generalization of the NAVE approach to the issue of value added by RCMs in climate-change experiments. It is my recommendation that this manuscript be accepted for publication with only minor revision.

General Minor Comments : 1) NAVE procedure applied to Climate-Change experiments: The NAVE procedure would seem to be equally applicable to climate change problems to help distinguish the impact of model formulation from the impact of resolution on RCM climate-change responses relative to those of its driving GCM. In the climate change context, two sets of RCM runs would need to be performed

- NAVE runs at the resolution of the driving global climate model (GCM) and the usual high-resolution runs used for downscaling GCM climate-change results. Consider a typical time-slice experiment over a CORDEX domain performed at the end of the 20th and 21st centuries. For a given climate index (eg screen-level temperature, precipitation, extremes etc.), one could construct the three climate-change responses: $R\_GCM(X) = GCM\_21st(X) - GCM\_20th(X)$ $R\_NAVE(X) = NAVE\_21st(X) - NAVE\_20th(X)$ $R\_RCM(X) = RCM\_21st(X) - RCM\_20th(X)$, where each term on the right is a time (and/or ensemble) average at a given spatial location "X". In the above, $R\_NAVE(X)$ represents the climate-change signal associated with model formulation differences between the RCM and GCM. As for the authors' present-day analysis, the potential for value added due to the response associated with resolution changes may be expressed as: $R\_RES(X) = R\_RCM(X) - R\_NAVE(X)$.

The NAVE analysis allows the decomposition: $R\_RCM = R\_NAVE(X) + R\_RES(X)$

Given $R\_RES(X)$, and $R\_NAVE$ one can ask interesting questions like: - Where is $R\_RES(X)$ significant in the RCM domain? - Do these locations correlate well with where the authors found downscaling improvement in their NAVE analysis of reanalysis driven RCMs? - Where is $R\_RCM$ appreciably different from $R\_NAVE$? The appreciable difference analysis presented in Section 5 of Scinocca et al. 2015 (JClim p. 17-35) would seem like an ideal approach to address this question. In locations where there exists an appreciable difference, there exists the potential for added value. However, where there is no appreciable difference, there can be no added value - irrespective of how one chooses to define added value. This is in line with the authors' stated goals (ll.116-118). Clearly such climate-change questions are outside the authors' present study but, they may want to discuss this potential application of the NAVE approach for future investigation.

Response: We really appreciate such detailed description on how the NAVE approach can be used for climate change projections and completely agree. Now, we also use the abbreviation "NAVE" in the study. Actually, we've already completed

downscaling of two global models (RCP8.5) over Africa at their native resolution, in addition to the standard 0.44deg CORDEX resolution. The first results were presented at EGU2019 (https://meetingorganizer.copernicus.org/EGU2019/EGU2019-7631.pdf) and at ICRC-CORDEX2019 (http://icrc-cordex2019.cordex.org/wp-content/uploads/sites/2/2019/11/AbstractBook_20191114.pdf, A1-P-38). A paper is in preparation. We added a paragraph in "Summary and Conclusion"

"Moreover, the same NAVE framework can be used for quantifying the added value in RCM-based future climate projections. For this, one needs to downscale GCMs at their native resolution in addition to the standard CORDEX resolutions (25 or 50km). The RCM projections at the native GCM resolution serve as the NAVE in the climate change context. A potential caveat, already mentioned in our study, is that RCMs are generally developed and tuned to operate at resolution of tens of km. "Downscaling" a GCM at its native resolution, for example 150 or 200km, may lead to artefacts related to a lack of RCM retuning for coarser resolution. Nerveless, more and more GCMs, for example in CMIP6, have resolution finer than 100km that allows application of the NAVE."

2) Interpretation of the NAVE: It is assumed here that differences in the NAVE and driving model results arise from differences the RCM and GCM model formulation. This would be strictly true only if the RCM were also run in a global mode. The one-way nesting approach introduces a number of potential artifacts which are most acute for large RCM domains and applications that do not use interior (or spectral) nudging - both of which are the case for the authors' present study (eg Section 2 of Scinocca et al. 2015 JClim p. 17-35). The authors should acknowledge this issue when introducing the NAVE.

Response: It's a very relevant comment as we missed this point. We agree that the one-way nesting approach is also a source of the difference between a RCM and its driving GCM. From our point of view, without spectral nudging, this source is mostly related to RCM domain configuration. If spectral nudging is not used, as in our NAVE

simulations, and a RCM develops its own climatology, the difference between the RCM and GCM is basically defined by RCM physical formulation and domain configuration. If spectral nudging is used, technical aspects of the nudging (e.g. which wavelengths should be nudged and at what altitudes) also contribute to the difference by reducing it. We added explanations in 2.2 Experiment design.

"The difference between a RCM and its driving GCM can, in general, be attributed to three sources, namely: i) different resolution, ii) different physical formulation and iii) artifacts of the one-way nesting approach including size of the RCM domain and application of spectral nudging (e.g. Scinocca et al., 2016). The RCA4 0.88° simulations and the HCLIM-ALADIN 100km one represent a slight upscaling of ERAINT (about 0.7° or about 77km at the Equator) and we refer to them as "no added value experiment" (NAVE). No resolution-dependent added value of the RCMs is expected for these simulations and all differences between the RCMs and their driving ERAINT are attributed to different physical formulations and to the artifacts of the one way nesting. Spectral nudging is not used in our experiment and the one way nesting term is basically reduced to domain configuration. In contrast, if spectral nudging is used, technical aspects of the nudging (e.g. which wavelengths should be nudged and at what altitudes) also contribute to the one way nesting term. In practice, it is not straightforward (if possible at all) to separate the impact of different physical formulation and artifacts of the one-way nesting approach. Hereafter, we use "RCM formulation" as a term that includes both RCM physical formulation and domain-dependent RCM configuration (e.g. size of the full domain)."

3) RCM model tuning: ll.183-185 "We note that in general, both regional models - RCA and HCLIM-ALADIN were developed to operate at a range of 10-50km resolution and their performance at 100 and 200km may not be optimal." This is a non-trivial point, given the philosophy of the authors' NAVE approach. Where there is systematic improvement of NAVE biases with increased resolution, the authors interpret this as a systematic increase in added value. However, The poorer results at the coarser

resolutions may also be related to a lack of model retuning at these non-standard resolutions. Very few physical parameterizations are automatically scale dependent and an adjustment of their free parameters with changing spatial resolution should in principle be performed. Retuning the RCMs at each spatial resolution would represent a significant undertaking and these added degrees of freedom would complicate the main point made in this study. Consequently, I would recommend that this issue be addressed by simply having it raised as a caveat.

Response: We completely agree with this comment and added some explanation on this potential caveat (see also response to Comment 3, 1st reviewer)

"We note that in general, both regional models - RCA and HCLIM-ALADIN were developed to operate at a range of tens of km resolution and their performance at 100 and especially at 200km may not be optimal. A potential caveat here is that very few RCM physical parameterisations are automatically scaled at very coarse resolution. Thus, results at the coarser resolutions may be partly related to the lack of model retuning. However, such coarse-resolution simulations are a useful supplement to simulations at a RCM comfortable resolution zone and help us to understand RCM behaviour without additional, resolution-dependent tuning."

4) Interior nudging: In downscaling reanalysis products, the authors chose not to employ any constraints on the interior RCM solution such as spectral nudging (ll.185-186). In focusing on such evaluation experiments, one could argue that it is more appropriate to use spectral nudging to constrain the large scales to obtain the best downscaled results in their study. Any upscale influence produced by the RCM would serve only to degrade the large scale flow as it is well observed and represented in reanalysis produces. By not constraining the RCM in this way, the authors leave open the possibility that locations of large biases in their high-resolution RCM results are due to the downscaling of the wrong large-scale flow rather than a lack of intrinsic added value. For more detail see Section 2 of Scinocca et al. 2015 (JClim p. 7-35).

Response: This is a reasonable comment. The CORDEX-Africa RCMs do not use the spectral nudging (see e.g. Nikulin et al. 2012) and here we follow the same approach for downscaling over Africa. A potential caveat for applying spectral nudging in the tropics was also shortly touched in Nikulin et al. 2012.

"With respect to spectral nudging of an RCM solution toward the driving data at large wavelengths (von Storch et al. 2000), this technique is well established for midlatitude regions, with some theoretical understanding of which wavelengths should be nudged and at what altitudes (Alexandru et al. 2009). This is not the case in the tropics, and it may be more difficult to formulate given the stronger role of surface forcing and mul-tiscale convection in driving large-scale circulations. We therefore chose to preclude spectral nudging from the experimental design, pending further work in this area."

We reformulated a bit:

"All simulations are conducted without spectral nudging similar to the CORDEX-Africa RCMs (Nikulin et al., 2012) allowing the RCMs to develop its own climatology as much as possible." Detailed Minor Comments:

l.26 "Additionally to the two RCMs" perhaps change to "In addition to the two RCMs"

Response: changed

l.31 "the phase of the diurnal cycle is" perhaps change to "the phase of the diurnal cycle in precipitation is"

Response: changed

l.71 "However, added value from RCMs" should be changed to "However, perceived added value from RCMs" for the context of the sentence.

Response: changed

ll.141-147. It was unclear whether the difference between v1 and v4 was simply a change in a free parameter for an existing scheme or whether the difference was associated with a change in the equations of the scheme. The former might be considered "tuning" while the latter considered a "formulation" difference.

Response: The difference between v1 and v4 is indeed simply a change in a parameter and we completely agree that such change can not be considered as a formulation difference but a new parameter setting or a new configuration. We added additional explanations on the difference between v1 and v4 in "2.1.1 RCA4" and also made a number of changes in the manuscript (using "two different parameter settings" for example).

"RCA4 has three configurations used for CORDEX simulations that are available through ESGF. They are named (so called RCM version) as v1 (Europe, Arctic, Africa, Southeast Asia, Central and North America, ref), v2 (South Asia, ref) and v3 (South America, ref) and differ in some domain-specific re-tuning. In this study we also include a new configuration - v4. The RCA-v4 is based on RCA4-v1 but with a change in one parameter leading to reduced turbulent mixing in stable situations (especially momentum mixing)."

ll.176-178 It would be helpful to show these plots to see if the differences have any correlation with later results (perhaps in an appendix) - particularly the distribution of temperature differences.

Response: We've again looked at this additional sensitivity experiment and found that actually there are also some differences in precipitation, not only in temperature. We reformulated our findings accordingly. At the same time, it's only one simulation at one resolution by one RCM. We would prefer not focus on this single simulation in detail. A full set of simulations with the same full domain for all RCMs and resolutions is necessary for robust conclusions and we leave more in-depth detailed analysis to forthcoming studies.

We reformulated the respective paragraph:
"As mentioned above, larger size of the computational domain at coarser resolution in our experiment setup may have a potential impact on the results leading to larger IV developed by the RCMs and weaker constraints on the ERAINT forcing. As a simple test for domain-dependant RCM IV we perform an additional experiment with RCA4 at 0.88° resolution taking the full computational domain from the 1.76° RCA4 simulation. Indeed, for the 1981-2010 climatology, seasonal mean precipitation differences between the two experiments can reach up to 1.25 mm/day (up to 25%) at a few individual grid boxes, often at the edges of the tropical rain belt, although in general stay below 0.5 mm/day (not shown). Seasonal mean temperature also differs with up to 1.25°C regionally (not shown). We do not focus on this single additional sensitivity experiment in the study. A full set of simulations with the same full domain for all RCMs and resolutions is necessary for robust conclusions. " ll.260-262 Fig 2b-p. It was often hard to associate the location of a particular bias with the full field in panel a. Expressing the bias as a percentage difference from the full field would be helpful in the West and central regions. However, where there is weak precipitation in the reference/obs data this may be problematic.

Response: It is a common problem for showing relative precipitation biases in the tropics when small reference values at the edge of the tropical rain belt or in dry regions lead to artificially excessive relative bias. One solution is to apply a filter, for example, to show only regions where reference precipitation is more than 1 mm/day. However, based on our experience such an approach does not always lead to better visualisation. Showing absolute biases is pretty common for model evaluation studies in Africa and we prefer to keep the absolute bias in Figs. 2 and 3. Additionally, following a comment of the 1st reviewer we mask by white all biases less than 0.5 mm/day and hope the visibility is better now.

ll.350-352 Fig 4. It would be better to use the colour red for the reference GPCC7 curves in this figure. I had difficulty seeing the GPCC7 curves in a number of the model result panels in columns 2-4.

Response: We changed the colour to red for the reference GPCC7 curves in Fig. 4. We also deleted the CRU and UDEL datasets from Fig. 4 as they simply coincide with GPCC7 and do not provide any useful information.

ll.558-560 "In general, model formulation related improvements cannot be considered as an added value of downscaling as such improvements are strongly model dependent and cannot be generalised." Also, such formulations could in principle be used in global models and so obviate the need for the RCM.

Response: We agree and added:

"However, such formulation-related and region-specific improvements from RCMs could in principle be also used in GCMs."

---

## Author Response (AR2)

Dear Editor,

Thank you for your suggestion on further improving the manuscript. We have now incorporated your suggestions into the current version of the manuscript, with some additional minor corrections for some words (e.g. tenses of verb) and unclear terms (e.g. ESGF). The revised manuscript in pdf format is now uploaded in the ESD submission system.

In the meantime, a point-by-point reply to your comments, as well as a marked-up version are provided at the end of this letter.

Thank you very much for your time helping with improving the manuscript, we look forward to your further advice in this review.

Best regards
Minchao Wu (on behalf of all co-authors)
* * *
**Comments from the Editor:**

**1. Page 4, line 79. I feel that rather than "wrong reason", "different reason" is a better phrase. There is nothing "wrong" about improvement in RCMs results due to better model formulation**.
Response: thanks for the suggestion, it is revised.

**2. Page 5, line 92. Please consider replacing ", thus" with "and therefore".**
Response: thanks for the suggestion, it is revised.

**3. Page 7, lines 130-131. Please consider replacing "that the CORDEX variable list only defines three pressure levels" to "that the CORDEX requires atmospheric variables at three pressure levels"**
Response: thanks for the suggestion, it is revised.

**4. Page 7, lines 136-138. Please consider replacing**
**"This deficiency of the RCMs is related to the convective parameterization used and a specific convection scheme, as for example the Kain–Fritsch (KF), may outperform others, producing a more realistic diurnal cycle (Nikulin et al., 2012)."**
**to**
**"This deficiency of the RCMs is related to the convective parameterization used and some convection schemes, for example the Kain–Fritsch (KF), may outperform others, producing a more realistic diurnal cycle (Nikulin et al., 2012)."**
Response: thanks for the suggestion, it is revised.

**5. Page 9, line 188. Please consider replacing "We need to note" to "We note".**
Response: thanks for the suggestion, it is revised.

**6. Page 10, lines 203-205. Please consider rewording the following sentence and provide a bit more info from**
**"However, an unnecessary large full domain for higher resolution simulations is a caveat leading to larger RCM internal variability (IV) and a higher computational demand at finer resolutions."**
**to**
**"However, an unnecessary large full domain for higher resolution simulations leads to larger RCM internal variability (IV) COMPARED TO .... and a higher computational demand at finer resolutions."**
Response: thanks for the suggestion, indeed, we need to improve the clarity here, now it reads:

"However, an unnecessary large full domain for resolutions finer than 200km (i.e. 100, 50 and 25km) leads to larger RCM internal variability (IV) compared to simulations at the same resolutions but with a minimum size full domain. Computational demands at the finer resolutions are also higher in the case of the large full domain."

**7. Page 10, line 211. Please consider replacing "although we should note that" with "although we note that".**

Response: thanks for the suggestion, it is revised.

**8. Page 12, line 243. Please consider replacing "automatically scaled at very coarse resolution" with "automatically scaled to run at very coarse resolution".**

Response: thanks for the suggestion, it is revised.

**9. Page 15, line 285. Please consider replacing "We also need to note that" with "We also note that".**

Response: thanks for the suggestion, it is revised.

**10. Finally, please make the colour for ERAINT consistent across Figures 4, 5, and 6. Right now ERAINT is represented by colour black in Figure 4 and colour red in Figures 5 and 6. This is somewhat confusing.**

Response: thanks for the suggestion, it is a bit confusing indeed. Color for ERAINT in Figure 4 is changed to color red and is now consistent with the ones in Figures 5&6.

[revised manuscript text omitted]